# NEMF mutations in mice illustrate how Importin-β specific nuclear transport defects recapitulate neurodegenerative disease hallmarks

Jonathan Plessis-Belair[1,2], Kathryn Ravano[1,2], Ellen Han[1,2], Aubrey Janniello[1,2], Catalina Molina[1,2], Roger B. Sher[1,2]*

**1** Department of Neurobiology and Behavior, Stony Brook University, Stony Brook, New York, United States of America, **2** Center for Nervous System Disorders, Stony Brook University, Stony Brook, New York, United States of America

* roger.sher@stonybrook.edu

**Data Availability Statement:** All Data are included in supplemental S1 Data.zip file.

## Abstract

Pathological disruption of Nucleocytoplasmic Transport (NCT), such as the mis-localization of nuclear pore complex proteins (Nups), nuclear transport receptors, Ran-GTPase, and Ran-GAP1, are seen in both animal models and in familial and sporadic forms of amyotrophic lateral sclerosis (ALS), frontal temporal dementia and frontal temporal lobar degeneration (FTD \FTLD), and Alzheimer's and Alzheimer's Related Dementias (AD/ADRD). However, the question of whether these alterations represent a primary cause, or a downstream consequence of disease is unclear, and what upstream factors may account for these defects are unknown. Here, we report four key findings that shed light on the upstream causal role of Importin-β-specific nuclear transport defects in disease onset. First, taking advantage of two novel mouse models of NEMF neurodegeneration ($Nemf^{R86S}$ and $Nemf^{R487G}$) that recapitulate many cellular and biochemical aspects of neurodegenerative diseases, we find an Importin-β-specific nuclear import block. Second, we observe cytoplasmic mis-localization and aggregation of multiple proteins implicated in the pathogenesis of ALS/FTD and AD/ADRD, including TDP43, Importin-β, RanGap1, and Ran. These findings are further supported by a pathological interaction between Importin-β and the mutant NEMF$^{R86S}$ protein in cytoplasmic accumulations. Third, we identify similar transcriptional dysregulation in key genes associated with neurodegenerative disease. Lastly, we show that even transient pharmaceutical inhibition of Importin-β in both mouse and human neuronal and non-neuronal cells induces key proteinopathies and transcriptional alterations seen in our mouse models and in neurodegeneration. Our convergent results between mouse and human neuronal and non-neuronal cellular biology provide mechanistic evidence that many of the mis-localized proteins and dysregulated transcriptional events seen in multiple neurodegenerative diseases may in fact arise primarily from a primary upstream defect in Importin-β nuclear import. These findings have critical implications for investigating how sporadic forms of neurodegeneration may arise from presently unidentified genetic and environmental perturbations in Importin-β function.

**Funding:** This work was supported by startup funds from Stony Brook University (63845 to RS), the Hartman Center for Parkinson's Research at Stony Brook University (64249 to RS), the Robert Packard Center for ALS Research at Johns Hopkins (2003084704 to RS), and the National Institutes of Health (R01AG079898 to RS). The funders had no role in study design, data collection and analysis, decision to publish, or preparation of the manuscript.

**Competing interests:** The authors have declared that no competing interests exist.

## Author summary

Neurodegenerative diseases are common disorders of aging, resulting in both motor and cognitive defects and premature death. Nucleocytoplasmic transport defects have been implicated in various forms of neurodegeneration including Alzheimer's and Related Dementias and Amyotrophic Lateral Sclerosis and Frontotemporal Dementia. Here, we describe nuclear transport defects in two mouse models with genetic mutations in Nuclear Export Mediator Factor (NEMF). We have found that these mice exhibit defects in nucleocytoplasmic transport, specifically in the nuclear import of proteins which are regulated through the Importin-β pathway. In addition, the two mutations result in very different ages of onset and progression of both disease and transport defects, mimicking the variation seen in genetic forms of human neurodegenerative diseases. They also recapitulate common pathologies and transcriptional alterations found in patients and are therefore a unique animal model of neurodegeneration. These findings are important because they point towards a potentially common pathway for disease development between rarer genetic and common sporadic forms of neurodegeneration.

## Introduction

Nucleocytoplasmic transport (NCT) defects underlie several neurodegenerative disorders, including amyotrophic lateral sclerosis (ALS), frontal temporal dementia and frontal temporal lobar degeneration (FTD\FTLD), and Alzheimer's and Alzheimer's Related Dementias (AD/ADRD) [1–3]. Here we show that mutations in Nuclear Export Mediator Factor (NEMF) in mice result in Importin-β-specific nuclear import defects, cytoplasmic protein mis-localization, and recapitulate phenotypic and transcriptional alterations found in neurodegenerative disorders. We further show that transient pharmaceutical inhibition of Importin-β induces key proteinopathies and transcriptional alterations seen in neurodegenerative diseases.

NCT involves the regulated trafficking of proteins and RNA between the nucleus and the cytoplasm through the nuclear pore complex (NPC) [4]. Smaller molecules (<40kDa) can freely migrate through the nuclear pore, whereas larger molecules require active transport mediated by nuclear transport receptors (NTRs, also known as karyopherins) such as importins and exportins [5]. During nuclear import, nuclear import receptors (NIRs), such as Importin-β, directly bind their cargo in the cytoplasm or utilize importin-α as an adaptor to bind cargo proteins bearing a classical nuclear localization signal (cNLS) [6, 7]. Other NIRs, such as Transportin-1, recognize different nuclear localization signals, such as a Proline-Tyrosine NLS (PY-NLS), for transport into the nucleus [8].

Pathological disruption of NCT, such as the mis-localization of Nuclear Pore Complex proteins (Nups), NTRs, and Ran-GTPase, are seen in both animal models and in familial and sporadic forms of ALS, FTD, and AD/ADRD [9–16]. In addition, pathological cytoplasmic protein aggregates mark the progression of most neurodegenerative diseases [17, 18]. One of the most frequently found protein aggregates comprises TAR-DNA binding protein-43 (TDP-43), normally a nuclear protein that, in disease, mis-localizes and aggregates in the cytoplasm [19]. Cytoplasmic aggregation and loss of nuclear TDP-43 are observed in post-mortem neurons and glia in 97% of cases in ALS, 40% of cases in FTD, and in many cases of AD/ADRD [20].

Amino acid substitutions in NEMF have been identified in multiple human patients exhibiting severe neurodevelopmental disorders [21–23]. However, the mechanisms through which these NEMF variants result in disease are unknown. While NEMF was initially associated with

nuclear transport in *Drosophila* (Caliban) [24], its canonical role has more recently been elucidated in the context of ribosome quality control (RQC). The RQC is recruited to stalled translation events [25–29], where NEMF plays a crucial role in targeting partially translated nascent chain polypeptides (NCPs) on the 60S ribosome for proteasomal degradation [24,28,30–33]. NEMF's interaction with the 60S ribosome facilitates the recruitment and stabilization of the E3 ubiquitin Ligase Listerin (LTN1), promoting the ubiquitination of NCPs [26,28,30,31,34,35]. Additionally, NEMF can generate c-terminal alanine (and threonine) tails on NCPs (CATylation) to expose amino acids in the ribosomal exit tunnel for subsequent ubiquitination by LTN1 [34,36,37]. These stalled translational events are most commonly a result of premature polyadenylation, where a poly (A) tail is inserted into the open reading frame resulting in the translation of lysine-AAA codons [26,28,29,38–40]. Notably, these so-called "nonstop" lysine-rich polypeptides that escape degradation by the RQC, as well as other defective ribosomal products (DRiPs), will be transported into the nucleus and will accumulate into the nucleolus before degradation [25,41]. Furthermore, Importin-β has been demonstrated to bind to translating NCPs targeted to the nucleus, suggesting that protein quality control and nuclear import mechanisms are more directly interconnected [42].

Mutations in both *Listerin* and *Nemf* in mice have been associated with neurodegenerative diseases [21,43]. Previously, Martin *et al.* [21] described two mouse models with different N-ethyl N-nitrosourea (ENU)-induced missense mutations in *Nemf* which develop progressive motor neuron degeneration. These two mouse lines carrying homozygous mutations in *Nemf*[R86S/R86S] and *Nemf*[R487G/R487G], henceforth *Nemf*[R86S] and *Nemf*[R487G], show progressive neuromuscular degeneration correlated with progressive loss of neuromuscular junction (NMJ) occupancy and axonal degeneration, with the *Nemf*[R86S] mice showing a more early-onset and severe phenotype. However, the exact nature of how these singe nucleotide polymorphisms result in a motor neuron disease and neurodegeneration remains unclear. In this study, we have demonstrated that the mutant mouse model, *Nemf*[R86S], exhibits impaired nuclear import of proteins containing canonical nuclear localization signals (cNLS), consequently leading to defects in nucleocytoplasmic transport both *in vitro* and *in vivo*. We have identified a specific defect in Importin-β mediated transport that manifests as both nuclear loss and cytoplasmic gain of NEMF, Importin-β, and TDP-43, along with a collapse of the Ran gradient. This dysregulation is further associated with the altered expression of transcripts associated with neurodegeneration, specifically *Stmn2*. Remarkably, the downregulation of *Stmn2* observed in our *Nemf* mutant-mouse model can be reproduced through the transient induction of a nuclear import block in both murine and human cells using a small molecular antagonist of Importin-β, importazole. Our study reveals that there is a causative role for an Importin-β nuclear import deficiency upstream of the multiple proteinopathies and transcriptional pathologies found in neurodegenerative disease.

## Results

### Motor Neuron Degeneration is associated with Progressive Nuclear loss of mutant NEMF in Lumbar Spinal Cord

To investigate the mechanism by which the missense *Nemf* mutation in mice led to an age-dependent neurodegenerative phenotype associated with motor deficits [21], we immunostained transverse lumbar spinal cords of wild type and *Nemf*[R86S] homozygous mutant mice at 21 days (*Nemf*[R86S] median lifespan: 20 days) [21]. Lumbar spinal motor neurons were identified by ChAT-positive cells located in the ventral horn. In wild type (WT) mice, NEMF was found to be predominantly nuclear with diffuse cytoplasmic staining (Fig 1A and 1B). *Nemf*[R86S] mice exhibited nuclear loss of NEMF with a significant cytoplasmic accumulation in

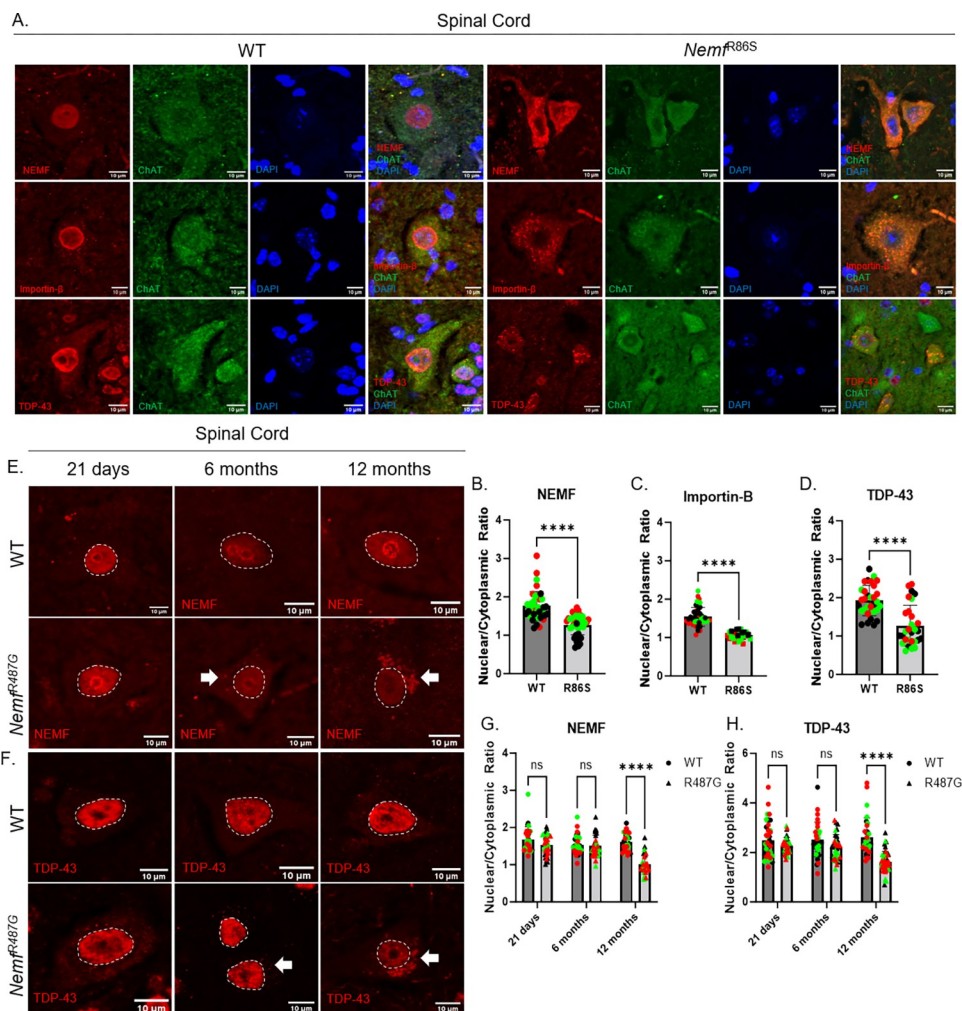

**Fig 1. Motor Neuron Degeneration is associated with Progressive Nuclear loss of mutant NEMF in Lumbar Spinal Cord.** A) Lumbar spinal cords were isolated from 21-day old Wild Type and *Nemf*[R86S] mice. Motor neurons in the ventral horn were immunostained for the nucleus (DAPI, blue), a motor neuron marker (ChAT, green) and NEMF (top row red), Importin-β (middle row red), or TDP-43 (bottom row red). B-D) Nuclear/Cytoplasmic ratios of proteins in A. Data analyzed by unpaired two-tailed t-test. (E-F) Immunostaining of NEMF and TDP-43 (red) in WT, and Nemf[R487G] lumbar spinal cord motor neurons at 21 days, 6 months and 12 months. G-H) Nuclear/Cytoplasmic ratios of protein of interests in WT and Nemf[R487G]. Data analyzed by two-way ANOVA with Šídák's multiple comparisons test. Individual colors in plots represents one animal. Arrow indicates cytoplasmic puncta. Scale bars are 10μm. (n = 30–44 cells) (ns p>0.05, **** p<0.0001).

the somata of ChAT positive motor neurons (Fig 1A and 1B). However, this phenotypic pathology is not exclusive to spinal motor neuron as staining in the primary motor cortex also displays a similar pattern of mis-localization (S1 Fig). Additionally, cytoplasmic mis-localization was also observed for Importin-β and TDP-43 (Fig 1A, 1C and 1D). Interestingly, the loss of nuclear localization of NEMF and TDP-43 was exclusive to neurons in the ventral horn, consistent with lamina IX [44] (S1 Fig).

To investigate an expanded temporal scale of the disease progression, we turned to the late onset *Nemf*[R487G] mouse that also displays a later-onset progressive neuromuscular disease with a milder disease phenotype [21]. The gain of NEMF and TDP-43 cytoplasmic inclusions occurred later in these *Nemf*[R487G] mice with their appearance beginning at approximately 6

months (Fig 1E and 1F). However, the significant nuclear loss of both NEMF and TDP-43 was observed later at 12 months, correlating more closely with disease onset (Fig 1E–1H). Thus, nuclear loss and cytoplasmic gain of mutant NEMF and TDP-43 occur during disease progression in mice with either of these NEMF mutant alleles.

## *Nemf*<sup>R86S</sup> exhibits specific Importin-β nuclear import defects

We generated WT and *Nemf*<sup>R86S</sup> mouse embryonic fibroblasts (MEFs) to investigate the mechanisms underlying the NEMF mutant phenotypes. To measure impacts on nuclear import, we used a turboGFP in-frame with a canonical SV-40 Importin-β nuclear localization signal (GFP-3X-NLS), which we exogenously expressed in *Nemf*<sup>R86S</sup> MEFs (S2 Fig). We found that GFP-3X-NLS appropriately translocated to the nucleus in WT MEFs, but predominantly remained in the cytoplasm in *Nemf*<sup>R86S</sup> MEFs (Fig 2A and 2B). A control plasmid with cytoplasmic turboGFP without an NLS showed no difference in localization and was found diffuse in both the nucleus and the cytoplasm, highlighting that the GFP-3X-NLS is being retained in the cytoplasm in *Nemf*<sup>R86S</sup> MEFs (Fig 2A and 2B). Importin-β and GFP-3X-NLS co-stains do not reveal any large sequestration of GFP-3X-NLS into cytoplasmic puncta in the *Nemf*<sup>R86S</sup> MEFs, but do reveal diffuse cytoplasmic colocalization (S2 Fig). To determine whether this Importin-ß defect observed is an indirect effect of RQC dysfunction, we knocked down the RQC E3 ubiquitin ligase *Ltn1* (S2 Fig). Knockdown of *Ltn1* reduces RQC capacity and prevents the RQC from targeting NCPs for degradation, resulting in the aggregation of its

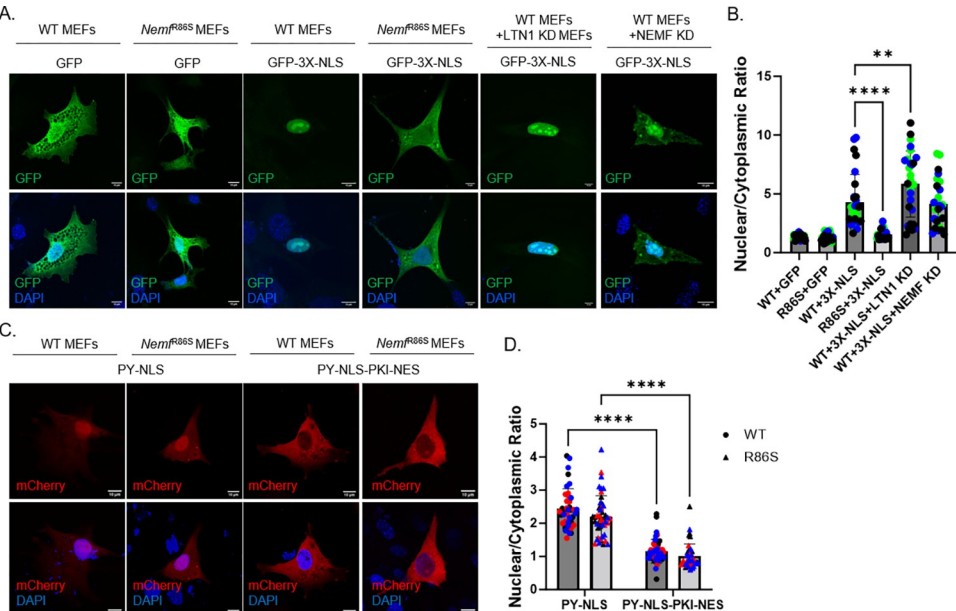

**Fig 2. *Nemf*<sup>R86S</sup> exhibits specific Importin-β nuclear import defects.** A) Expression of turbo GFP and turbo GFP inframe with three Canonical SV-40 Nuclear Localization Signal (3X-NLS) (green) in WT and *Nemf*<sup>R86S</sup> MEFs, and in LTN1 and NEMF siRNA treated WT MEFs. Nuclear stain is DAPI (blue). B) Nuclear/Cytoplasmic Ratios of GFP and GFP-3X-NLS in WT and *Nemf*<sup>R86S</sup> MEFs. Data Analyzed by one-way ANOVA with Tukey's multiple comparisons test (n = 28–32 cells) C) Expression of mCherry in frame with a Proline-Tyrosine Transportin-1 Nuclear Localization Signal (PY-NLS) and mCherry in frame with a PY-NLS and a pKI Exportin-1 Nuclear Export Signal (PY-NLS-PKI-NES) (red) in WT and *Nemf*<sup>R86S</sup> MEFs. Nuclear stain is DAPI (blue). D) Nuclear/Cytoplasmic Ratios of PY-NLS and PY-NLS-PKI-NES in WT and *Nemf*<sup>R86S</sup> MEFs. Data analyzed by two-way ANOVA with Šídák's multiple comparisons test. (n = 40–46) Individual colors in plots represent one trial. Scale bars are 10μm. (ns p>0.05, ** p<0.0001, **** p<0.0001).

substrates[32, 36, 45]. We observed no significant difference in the ability to localize GFP-3X-NLS in with *Ltn1* knockdown in WT MEFs (Fig 2A and 2B), highlighting that the GFP-3X-NLS is not an RQC substrate. Next, we knocked down NEMF (S2 Fig) to determine if this defective nuclear import is either a loss of function or a toxic gain of function phenotype from the mutant form. NEMF knockdown in the WT cells did not result in an overt quantitative nuclear import block, but did present with cytoplasmic inclusions (Fig 2A and 2B). NEMF knockdown in the *Nemf*^R86S only resulted in a 20% knockdown of transcript, but concurrently resulted in a large decrease in cell survival (S2 Fig), suggesting that NEMF knockdown has a significant toxic effect on these mutant cells.

We then expressed a Proline-Tyrosine (PY) Transportin-1 targeted nuclear localization signal in-frame with mCherry and observed no significant difference in nuclear import (Fig 2C and 2D), indicating that only active Importin-β import, but not active Transportin-1 import, is affected. Proper nuclear import of the PY-NLS reporter in both WT and *Nemf*^R86S allowed us to create a chimeric protein expressing both a PY-NLS and a pKI nuclear export signal (pKINES) in-frame with an mCherry. Thus, following the initial import into the nucleus, this pKINES-PY-NLS nuclear export reporter would be shuttled out into the cytoplasm and indeed, both the WT and mutant *Nemf*^R86S are able to properly export this reporter (Fig 2C and 2D). To confirm that this nuclear import reporter GFP-3X-NLS is Importin-β specific, we transiently treated these MEFs with an Importin-β small molecular inhibitor Importazole (IPZ) [46]. Indeed, inhibition of Importin-β with IPZ in WT cells resulted in the accumulation of the GFP-3X-NLS reporter in the cytoplasm but had no effect on the localization of GFP or the PY-NLS reporter (S2 Fig). Thus, mutant NEMF results in a nuclear transport defect that is specific to the Importin-β pathway, leaving transportin-1 mediated import and exportin-1 mediated export intact.

To further confirm that this defect is not a result of a passive nuclear import block, we digitonin-permeabilized these MEFs and measured the ability for 60-70kDa and 500kDa dextrans conjugated to FITC to passively diffuse into the nucleus. We observed no significant difference with 500kDa dextrans but observed a significantly greater nuclear localization of the 60-70kd dextrans in the *Nemf*^R86S MEFs, highlighting a potentially 'leaky' nuclear pore in the mutant cells (S2 Fig).

## RQC nonstop and poly (A) stalling reporters fail to localize to the nucleus in *Nemf*^R86S MEFs

RQC substrates that escape degradation have been previously described to be transported into the nucleus where they will then accumulate into the nucleolus [25]. This accumulation into the nucleus was dependent on the lysine-enriched 3' UTR, where poly-lysines tails were sufficient to cause nucleolar accumulation [25]. The homology of these poly-lysine tails and lysine enriched canonical SV-40 NLS sequences suggests that the transport into the nucleus may be Importin-β dependent. Here we show that nonstop stalling reporters that escape degradation fail to be transported into the nucleus and accumulate in the cytoplasm of *Nemf*^R86S MEFs (Fig 3A and 3B). The hydrophobic nature of the 3'UTR in the nonstop reporter (GFP-Nonstop) increased its propensity to aggregate in the *Nemf*^R86S MEFs, presenting as cytoplasmic puncta (Fig 3A). In comparison, the addition of a Poly-Lysine track (Poly-K (AAA)) in the C terminal also failed to properly localize into the nucleus in the *Nemf*^R86S MEFs but lacked the high presence of cytoplasmic puncta (Fig 3A and 3B). Both of these substrates were found to be sequestered in the cytoplasm with Importin-β in our mutant cells, whereas this cytoplasmic sequestration was absent when expressing the GFP-Stop reporters (Fig 3A). Immunoblotting for these GFP reporters revealed a partial reduction in the CATylation ability as indicated by

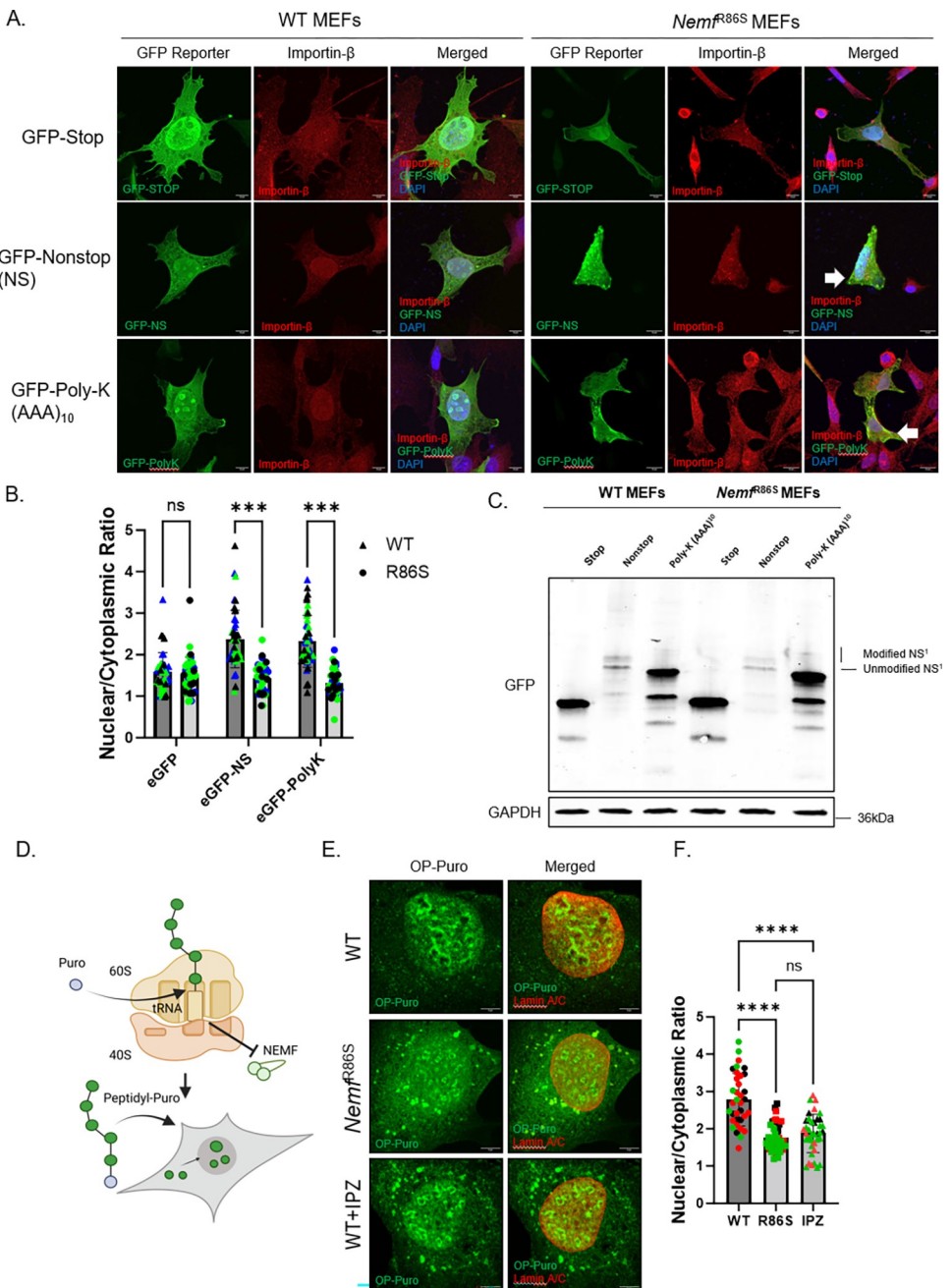

**Fig 3. RQC nonstop and poly (A) stalling reporters fail to localize to the nucleus in *Nemf*[R86S] MEFs** A) Expression of GFP-Stop, GFP-Nonstop, and GFP-PolyK-(AAA)$_{10}$ (green) co-stained with Importin-β (red) in WT and *Nemf*[R86S] MEFs. Nuclear stain is DAPI (blue). B) Nuclear/Cytoplasmic Ratios of GFP-Stop, GFP-Nonstop, and GFP-PolyK-(AAA)$_{10}$ in WT and *Nemf*[R86S] MEFs. Data Analyzed by two-way ANOVA with Šídák's multiple comparisons test. Scale bars are 10μm. (n = 39–46). C) Western Blot Analysis of GFP-Stop, GFP-Nonstop, and GFP-PolyK-(AAA)$_{10}$ expression in WT and *Nemf*[R86S] MEFs. D) Schematic of Puromycin (Puro) inclusion into translated nascent chains polypeptides (NCPs) resulting in the inhibition of NEMF and the release of the nascent chain into the cytoplasm. The NCPs are transiently imported into the nucleus and accumulate into nucleoli. E) Immunostaining of puromycylated NCPs (OP-Puro, green) and Lamin A/C as a nuclear marker (red) in WT, *Nemf*[R86S] MEFs, and IPZ-treated WT MEFs. F) Nuclear/Cytoplasmic Ratios of puromycylated NCPs in WT, *Nemf*[R86S] MEFs, and IPZ-treated WT MEFs. Data analyzed by ordinary one-way ANOVA with Tukey's multiple comparison test. Individual colors in plots represent one trial. Scale bars are 5μm. (n = 35–43 cells) (ns p>0.05, *** p<0.001, **** p<0.0001).

the reduction of modified nonstop substrates versus unmodified (Fig 3C), as previously described in yeast [21]. We then tested whether this cytoplasmic accumulation was RQC dependent or if it was an Importin-β specific defect. We utilized puromycin which induces translational inhibition by incorporation into the nascent chain polypeptide and subsequent release from the ribosome (Fig 3D). This incorporation and release prevents the dissociation of the 60S and 40S ribosome, preventing NEMF and other RQC components from binding to the ribosome and targeting these defective ribosomal products for degradation [30]. These puromycylated nascent chain polypeptides bypass RQC-mediated degradation and will also accumulate in the nucleolus [41]. Indeed, when the *Nemf*$^{R86S}$ MEFs were pulsed with puromycin (OP-Puro), we observed a reduction for these substrates to be properly transported into the nucleus compared to WT MEFs, resulting in their accumulation in the cytoplasm (Fig 3E and 3F). This cytoplasmic accumulation was also observed when WT MEFs were treated with an Importin-β antagonist IPZ [46], highlighting the role of Importin-β as both a nuclear import mediator and a cytoplasmic chaperone (Fig 3E and 3F).

### *In Situ* Proximity Ligation Assay (PLA) reveals cytoplasmic NEMF R86S interactions with Importin-β

Importin-β has been shown to form cytoplasmic granules under basal conditions as well as under stress [47]. It has been demonstrated that under conditions of stress such as neurodegenerative disease, both the abundance and size of these granules increases [47]. To identify a mechanism for this mutant *Nemf*$^{R86S}$ nuclear import defect, we co-stained for Importin-β and NEMF and observed an increase in the size, but not abundance of Importin-β granules (Fig 4A–4C). This increase in size was also correlated with an increase in the number of granules colocalizing with NEMF in the R86S cells (Fig 4D). This prompted us to investigate

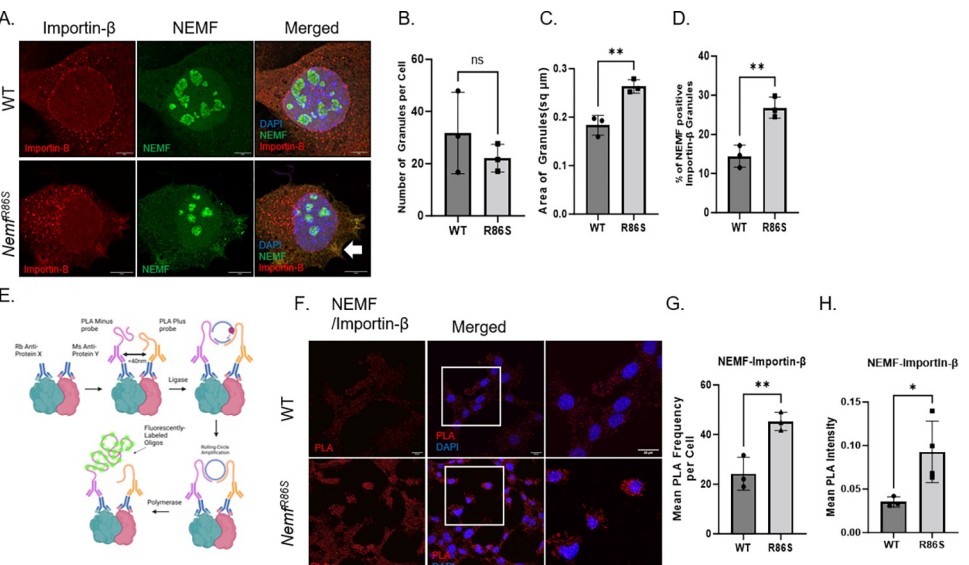

**Fig 4.** *In Situ* **Proximity Ligation Assay (PLA) reveals cytoplasmic NEMF R86S interaction with Importin-β.** A) Immunofluorescent staining of NEMF (green) and Importin-β (red) in WT and *Nemf*$^{R86S}$ MEFs. Nuclear stain is DAPI (blue). Scale bars are 5μm. B-D) Quantification of Importin-β granules, area of granules (square microns), percent of NEMF positive Importin-β granules (n = 3). E) Schematic of PLA adapted from Hegazy et al., 2020 [49]. F) PLA of NEMF and Importin-B (red). Nuclear stain is DAPI (blue). Scale bars are 20um. G-H) Quantification of Mean PLA frequency and PLA Intensity (n = 3). All data was analyzed by unpaired two-tailed t-test. (ns p>0.05, **p<0.01, *p<0.05, ****p<0.0001).

whether the mutant NEMF protein directly interacts with Importin-β in the cytoplasm. We utilized an in-situ proximity ligation assay using probes for NEMF and Importin-β and saw both an increase in the mean PLA frequency as well as the mean PLA intensity in the mutant $Nemf^{R86S}$ MEFs (Figs 4E–4H and S3). Thus, there is an increase in the frequency of interaction between Importin-β and NEMF R86S relative to that with Importin-β and WT NEMF protein. Whether these interactions are a result of direct binding of mutant NEMF protein to Importin-β, or a byproduct of sequestration in a larger complex is yet to be determined [48]. Interestingly, even in WT cells, there is some association of NEMF and Importin-β, indicating that NEMF may have some normal function in Importin-β biology.

## $Nemf^{R86S}$ MEFs display cytoplasmic aggregates

The identification of NCT defects in this mutant $Nemf^{R86S}$ model prompted us to investigate whether pathology was presented in the form of cytoplasmic aggregates. $Nemf^{R86S}$ MEFs displayed a smaller cell area and a significantly decreased growth rate (Doubling time-WT:17.32h, NEMF R86S: 26.20h) (S4 Fig). Furthermore, a puromycin incorporation assay showed no significant differences in the incorporation of puromycin, highlighting no greater differences in protein synthesis (S4 Fig). Immunostaining for NEMF, TDP-43, Ran, and Ran-GAP1 protein revealed cytoplasmic puncta (Fig 5A). This phenotype was not uniformly observed, as only a fraction of the cells (20–40%) exhibited this aggregated phenotype (S4 Fig). In addition to the canonical nuclear import function of Importin-β, it has been described to act as a cytoplasmic chaperone to prevent aggregation-prone proteins from misfolding and accumulating into insoluble aggregates [48]. The mis-localization into cytoplasmic puncta of NEMF, TDP-43, Ran, and RanGAP1 co-localized with Importin-β suggests that the primary defect of NEMF mutations is a direct dysfunction of Importin-β (Figs 5A and S4). AmyloGlo (Biosensis) labeling also revealed a significant increase in the amyloid-like signal in the cytoplasm in the mutant MEFs compared to the WT MEFs (S4 Fig). Treatment of these cells with Importin-β antagonist IPZ resulted in the lower cell survival in the $Nemf^{R86S}$ MEFs with ~48% cell survival at 40μM as compared to ~85% cell survival in WT MEFs (S4 Fig). Despite the mis-localization of these nuclear transport factors, factors such as Exportin-1 and Transportin-1 were not observed to be mis-localized or aggregated (S5 Fig). Furthermore, other factors such as NPM1, a nucleosome marker that co-localizes with NEMF in nucleolar puncta, were also not observed mis-localized in the cytoplasm (S5 Fig). Moreover, ribosomal subunits 40S (RPS6) and 60S (RPL3) were also not observed overtly mis-localized (S5 Fig). Various nucleoporins (Nup153, Nup98, and mAb414 NPC) were also observed to be sequestered within Importin-β cytoplasmic puncta and showed a reduction in nuclear signal (S5 Fig). As TDP-43 was found to be mis-localized, pathological TDP-43 was identified through phospho-TDP-43 immunostaining and was found to be greatly increased in mutant cells (~21% of cells) (Fig 5B and 5C). We then biochemically isolated TDP-43 in WT and $Nemf^{R86S}$ MEFs using sarkosyl insoluble fractionation [50] and found an increase in levels of insoluble TDP-43 in mutant cells (Fig 5D and 5E). Immunostaining of spinal motor neurons in WT and mutant $Nemf^{R86S}$ mice revealed cytoplasmic phospho-TDP-43 aggregation exclusive to these ChAT-positive motor neurons, with ~36% of these motor neurons displaying pTDP-43 cytoplasmic inclusions (Fig 5F and 5G). Subsequent staining of the late onset mutant $Nemf^{R487G}$ spinal motor neurons revealed some phospho-TDP-43 positive motor neurons at 6 months (~4%), with more motor neurons positive at 12 months (~18%) (S6 Fig).

Taken together, these findings demonstrate that NEMF mutations result in nuclear import defects and cause cytoplasmic protein aggregates, leading to the sequestration of nuclear transport factors and TDP-43 pathology.

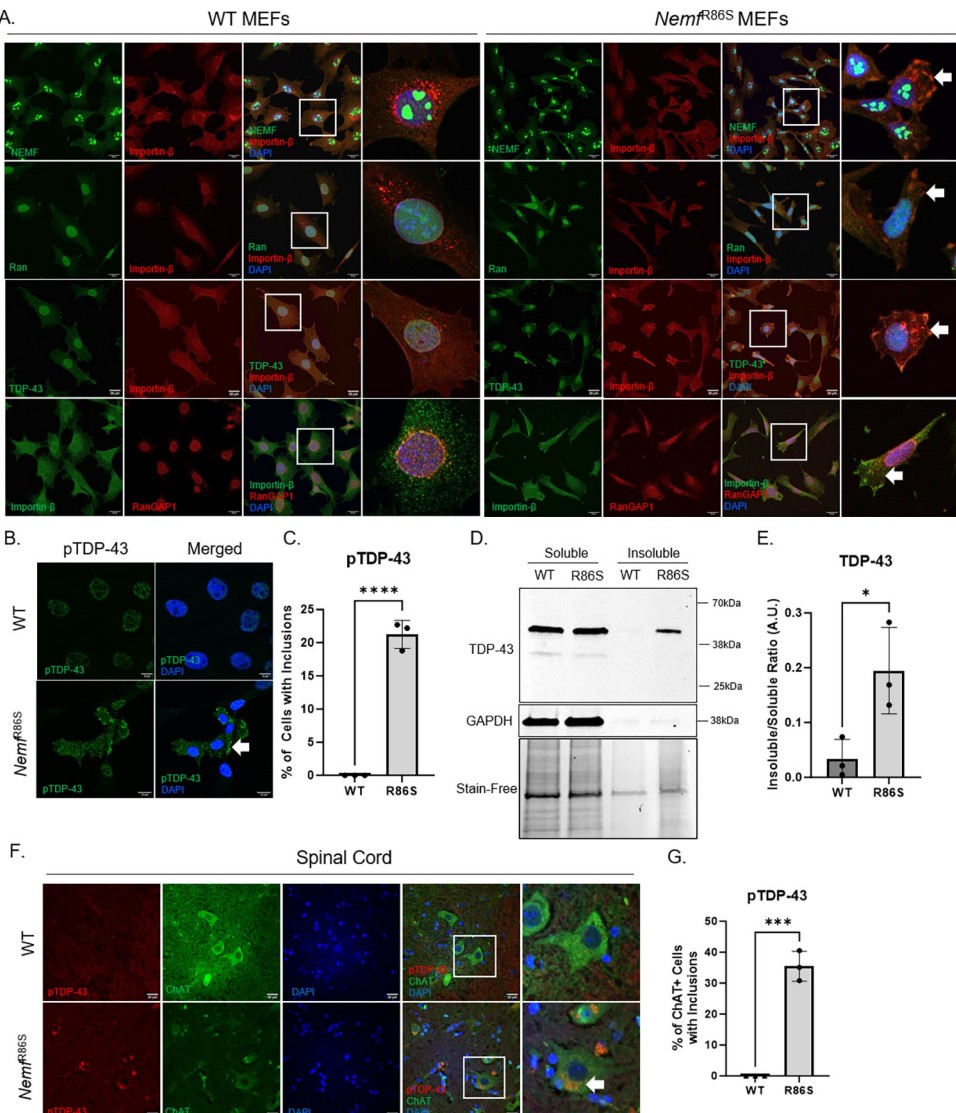

**Fig 5. *Nemf*<sup>R86S</sup> MEFs display cytoplasmic aggregates.** A) Immunofluorescent Staining of NEMF, Ran, TDP-43, and RanGAP1 with Importin-β in WT and *Nemf*<sup>R86S</sup> MEFs. Nuclei Labeled with DAPI (blue). Scale bars are 20μm. B) Immunofluorescent Staining of pTDP-43 in WT and *Nemf*<sup>R86S</sup> MEFs. Scale bars are 10μm. C) Percentage of cells with pTDP-43 cytoplasmic inclusions in WT and *Nemf*<sup>R86S</sup> MEFs (n = 3). D) Western Blot Analysis of sarkosyl soluble and insoluble TDP-43 in WT and *Nemf*<sup>R86S</sup> MEFs E) Ratio of Soluble versus Insoluble TDP-43 protein levels (n = 3). J) Immunofluorescent staining of phospho-TDP-43 (red), ChAT (green) in WT, and *Nemf*<sup>R86S</sup> lumbar spinal cord motor neurons at 21 days. Nuclei Labeled with DAPI (blue). Scale bars are 20μm. G) Percentage of ChAT+ cells with pTDP-43 cytoplasmic inclusions in WT and *Nemf*<sup>R86S</sup> mice (n = 3). Individual colors in plots represent one trial. Arrow indicates cytoplasmic puncta. (*p<0.05, ***p<0.001,****p<0.0001).

## *Nemf*<sup>R86S</sup> mice display altered expression of neurodegenerative disease related genes

The nuclear loss of TDP-43 prompted us to examine transcriptional profiles in this mutant model to determine whether there were similar transcriptional changes as those seen in neurodegeneration with TDP-43 dysfunction. Indeed, a subset of ALS- and AD-linked genes displayed altered expression in the mutant MEFs, with *Stmn2* and *Apoe* being highly down- and upregulated, respectively (Fig 6A and 6B). qPCR validation in MEFs shows that both

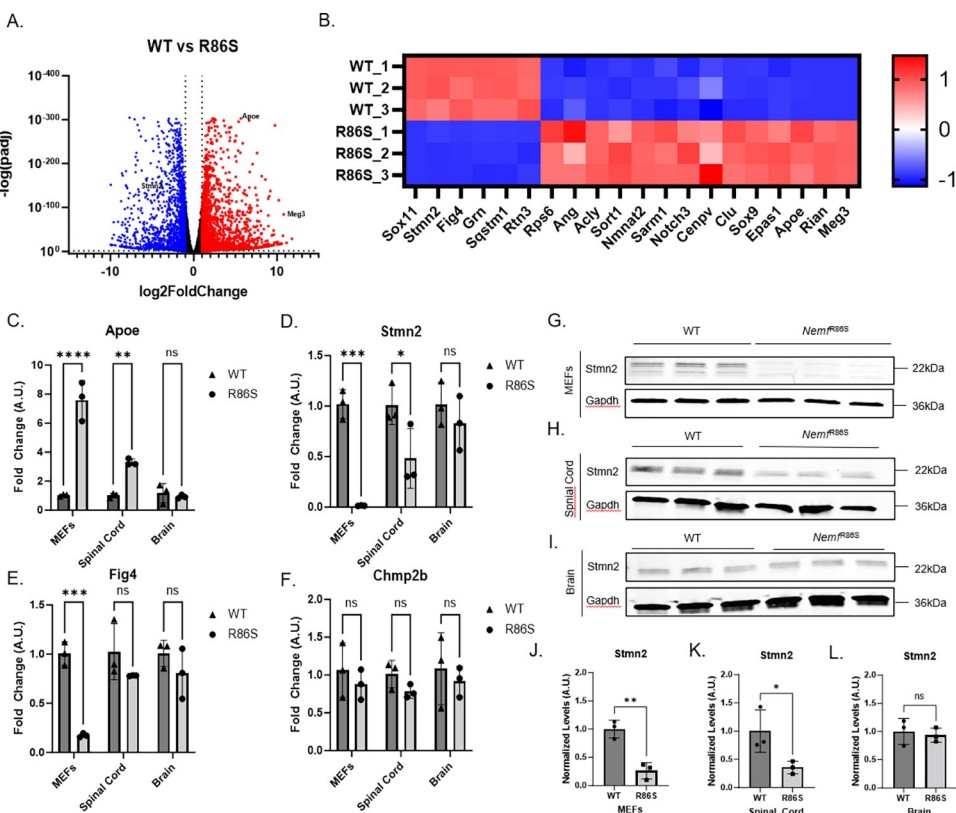

**Fig 6. *Nemf*[R86S] mice display altered expression of neurodegenerative disease related genes.** A) Volcano plot of upregulated (red) and downregulated (blue) genes of bulk RNA-seq in *Nemf*[R86S] MEFs relative to WT MEFs. (Log2foldchange threshold <-1, >1, p<0.05). B) Heatmap of significantly dysregulated genes commonly observed in neurodegeneration. (C-F) qPCR relative fold change of *Stmn2, Apoe, Chmp2b, and Fig 4* in MEFs, spinal cord, brain in WT and *Nemf*[R86S] mice. Data analyzed by two-way ANOVA with Šídák's multiple comparisons test. (n = 3). G) Western Blot Analysis of STMN2 protein levels in MEFs, Spinal Cord, and Brain. J-L) Quantification of STMN2 protein levels normalized to GAPDH levels. Data analyzed by unpaired two-tailed t-test. (n = 3) (ns p>0.05, *p<0.05, **, p<0.01, *** p<0.001, **** p<0.0001).

transcripts are significantly dysregulated in the mutant, with significant dysregulation also found in lumbar spinal cord tissue, but not in brain tissue (Fig 6C and 6D). Other notable genes found to be dysregulated in neurodegeneration and were differentially expressed in the mutants include *Fig 4*, *Bax*, and *Sort1* [51–53], which were significantly dysregulated *in vitro* but not *in vivo* (Figs 6E and S7). Another gene commonly dysregulated in neurodegeneration, *Chmp2b* [54–57], was downregulated in all tissue types, but was not significant (Fig 6F). Investigations into bonafide TDP-43 missplicing events revealed an increase in exon inclusion of *Sort1* and *Dnajc5* (S7 Fig), consistent with nuclear loss of TDP-43 [58,59]. Additionally, an mutant MEFs display upregulation of long noncoding RNA, *Meg3*, which has been implicated in necroptosis in Alzheimer's disease (Fig 6A and 6B) [60]. To determine neuronal-specific relevance of these changes, we isolated primary cortical neurons from WT and *Nemf*[R86S] mice and cultured them for 14 days prior to RNA extraction (S7 Fig). We observed a significant downregulation of *Stmn2* transcripts, but not *Apoe* (S7 Fig). To validate RNAseq data to qPCR data, simple linear regressions were used to model fold change in RNA-seq data and qPCR data, showing significant correlations in MEFs and spinal cord tissue, but not brain tissue (S8 Fig). Gene ontology and pathway analysis of the RNA-seq dataset of WT and mutant *Nemf*[R86S] MEFs revealed biological processes involved in cell adhesion, skeletal system

development, and negative regulation of cell proliferation (S7 Fig). An investigation into STMN2 protein levels revealed STMN2 protein loss *in vitro* and *in vivo*, and specifically in the spinal cord, but not the brain, consistent with RNA downregulation (Fig 6G–6L). A possible confound for the differences in the degree of dysregulation from *in vitro* to *in vivo* and between spinal cord and brain *in vivo*, could result from both the initial onset of pathology (i.e. spinal motor neurons vs brain) and the nature of the non-specificity of bulk tissue RNA extraction. Collectively, these observations, both *in vitro* and *in vivo*, indicate that a primary pathology triggered by canonical nuclear import defects results in TDP-43 pathology and downstream transcriptional dysregulation in sporadic forms of ALS-FTD and neurodegenerative disease.

## A transient nuclear import block recapitulates transcriptional downregulation of *Stmn2*

To investigate if impairing the Importin-β pathway itself recapitulates the phenotypes seen in the NEMF mutant, we targeted the nuclear import pathway through the small molecular inhibitor of Importin-β, IPZ [46]. A cell viability assay was used to determine sub-toxic levels of importazole (20uM) following a 48h treatment (S9 Fig). We found that transiently blocking Importin-β nuclear import results in the mis-localization of TDP-43 with the presence of cytoplasmic phospho-TDP-43 in ~15% of cells (Figs 7A, 7B and S9) and in the downregulation of *Stmn2* transcript, with no alteration of *Apoe* transcripts (Fig 7C and 7D). To confirm that *Stmn2* downregulation was in fact downstream of blocking nuclear import, and not simply a side effect of importazole treatment, we also treated MEFs with sub-toxic levels of ivermectin (IVM, 10uM, S9 Fig), a known Importin-β antagonist [61]. Using this transient nuclear import block, we observed a similar downregulation of *Stmn2* transcript, with no *Apoe* dysregulation (Fig 7C and 7D). The lack of *Apoe* dysregulation downstream of a nuclear import block suggests that *Apoe* upregulation may be an Importin-β-independent event or may have lower sensitivity to short-term transient Importin-β blockage than that seen *in vivo*. To further ensure that the downregulation observed was not a result of a decrease in RNA quality, as has previously been described to be a confounding factor [62], RNA quality was checked and confirmed to be of the highest quality, with RINe values ranging from 9.4–9.7 across all samples (S8 Fig). Following these observations, Importazole-treated RNA samples were sent for bulk RNA sequencing to determine which transcripts were similarly dysregulated to our *Nemf*$^{R86S}$ MEFs (Fig 7E). We found 20 commonly dysregulated genes in both conditions, with 15 of those genes similarly downregulated, and *Stmn2* being the only ALS-linked gene commonly downregulated (Fig 7F and 7G). A gene ontology analysis also described cell adhesion as being the most significant dysregulated pathway (similar to the *Nemf*$^{R86S}$ MEFs) with collagen fibril organization, extracellular matrix organization, and many cholesterol and lipid processes also observed as dysregulated pathways (S7 and S9 Figs). We then isolated primary cortical neurons from WT mice and cultured them for 14 days prior to RNA extraction (S9 Fig). These primary neurons were treated with IPZ for 48 hours prior to RNA extraction. Similar to *Nemf*$^{R86S}$ primary neurons (S7 Fig), we observed a significant downregulation of *Stmn2*, but not *Apoe* transcripts (S9 Fig).

Using *Apoe* and *Stmn2* dysregulation as a marker for transcriptional dysregulation, we targeted *Nemf* and *Ltn1* by siRNA knockdown in WT and *Nemf*$^{R86S}$ (S2 Fig) and saw that *Nemf* knockdown causes a small, but significant reduction in *Stmn2* transcript and an almost 3-fold increase in *Apoe* transcript, comparable to the mutant mouse model (Fig 7H and 7I). In comparison, *Ltn1* knockdown samples did not perturb *Stmn2* and *Apoe* transcriptional regulation (Fig 7H and 7I). The results indicate that *Apoe* dysregulation may be more related to NEMF impairment than to Importin-β impairment, while *Stmn2* seems to be more directly impaired by Importin-β.

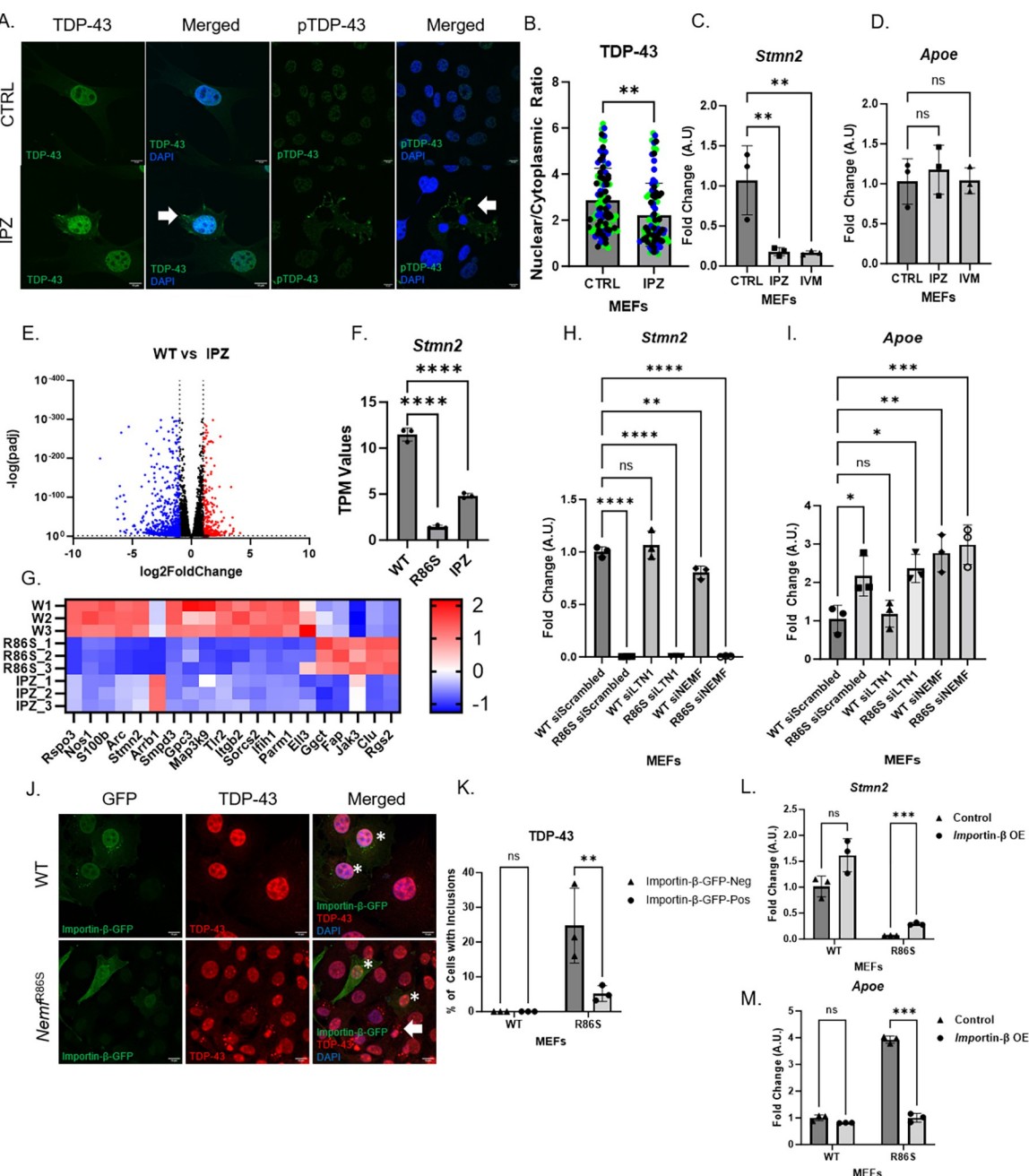

**Fig 7. A transient nuclear import block recapitulates transcriptional downregulation of *Stmn2*. A)** Immunofluorescent staining of TDP-43 and pTDP-43 (green) in WT and Importazole-treated (IPZ) MEFs. Nuclei Labeled with DAPI (blue). Scale bars are 10μm. **B)** Quantification of Nuclear/Cytoplasmic Ratio of TDP-43 (n = 92–99). Data analyzed by unpaired two-tailed t-test. **C-D)** qPCR relative fold change in DMSO Control and IPZ and Ivermectin (IVM) treated MEFs for indicated genes (n = 3). **E)** Volcano plot of upregulated (red) and downregulated (blue) genes of bulk RNA-seq in IPZ-treated WT MEFs relative to WT MEFs. (Log2foldchange threshold <-1, >1, p<0.05). **F)** *Stmn2* transcripts per million values for WT, *Nemf*$^{R86S}$, and IPZ-treated MEFs. **G)** Heatmap of significant commonly dysregulated genes in WT, *Nemf*$^{R86S}$, and IPZ-treated MEFs. (n = 3) **H-I)** qPCR relative fold change of *Stmn2* or *Apoe* in WT or *Nemf*$^{R86S}$ MEFs treated with *Ltn1* or *Nemf* siRNAs (n = 3). Data analyzed by one-way ANOVA with Tukey's multiple comparison test. **J)** Exogenous expression of Importin-β-GFP (green) co-stained with TDP-43 (red) in WT and *Nemf*$^{R86S}$ MEFs. Nuclei Labeled with DAPI (blue). Asterisks indicate GFP-positive cells. **K)** Percentage of cells with pTDP-43 cytoplasmic inclusions in WT and *Nemf*$^{R86S}$ MEFs. Data Analyzed by two-way ANOVA with Šídák's multiple comparisons test (n = 3) **L-M)** qPCR relative fold change of *Stmn2* or *Apoe* in WT or *Nemf*$^{R86S}$ MEFs with or without overexpression of Importin-β (Importin-β OE). Data Analyzed by two-way ANOVA with Šídák's multiple comparisons test (n = 3). Arrows indicate cytoplasmic inclusions. Individual colors in plots represent one trial. (ns p>0.05, *p<0.05, **, p<0.01, *** p<0.001, **** p<0.0001).

To confirm that this transcriptional dysregulation can be recapitulated in human models, we treated SK-N-MC neuroblastoma cells with importazole for 48h and observed an increase in the cytoplasmic mis-localization of TDP-43 with the presence of cytoplasmic phospho-TDP-43 (S10 Fig). This TDP-43 mis-localization was accompanied by a ~50% reduction in *STMN2* transcripts (S10 Fig). We then proceeded to knock down *NEMF* (SDCCAG1) and *LTN1* (RNF160) in HEK293 and SK-N-MC cells (S10 Fig) and observed a similar downregulation of full-length *STMN2* transcripts in si*NEMF* treated SK-N-MC cells, but not si*LTN1* treated samples (S10 Fig) si*NEMF* treated HEK293 cells did show a trend towards lower STMN2 transcripts, but was not significant (S10 Fig). We also observed an increase in *APOE* transcripts in si*NEMF* treated HEK293 cells (S10 Fig), but not in si*LTN1* treated HEK293 cells or in both treatments in SK-N-MC cells (S10 Fig). To further elucidate how nuclear import mechanisms can dysregulate *STMN2* transcriptional regulation, we targeted nuclear import pathways through siRNA knockdown (*Importin-β* (KPNB), *Importin-α* (KPNA), and *Transportin-1 (TPNO1)*) in HEK293 cells (S10 Fig) and *Importin-β* in SK-N-MC cells (S10 Fig). Interestingly, we see a downregulation of *STMN2* transcripts when Importin-β nuclear import is inhibited either by *Importin-β* siRNA-treated or IPZ-treated samples in both HEK293 and SK-N-MC cells (S10 Fig). Furthermore, only knockdown of *Importin-α* in HEK293 resulted in a significant upregulation of *APOE* (S10 Fig).

Lastly, Importin-β upregulation has been demonstrated to rescue hallmarks of TDP-43 proteinopathies and has demonstrated strong therapeutic potential in neurodegenerative proteinopathies [63, 64]. Using these strategies, we upregulated Importin-β expression through transient transfection to determine if the upregulation of this protein can result both the dissolution and disaggregation of TDP-43 in these *Nemf*^R86S MEFs. Indeed, overexpression of Importin-β-GFP resulted in a decrease in the presence of cytoplasmic TDP-43 inclusions in these Importin-β-GFP-positive cells (Fig 7J and 7K). Furthermore, overexpression of Importin-β-GFP was able to partially restore depleted levels of *Stmn2* in these *Nemf*^R86S MEFs, and almost completely restored levels of *Apoe* (Fig 7K and 7L).

Altogether, our results show that targeting nuclear import pathways can recapitulate *Stmn2* downregulation in both mice and human cell lines, whereas *Apoe* dysregulation is non-neuronal specific and may be more specific to NEMF impairment than Importin-β.

## Discussion

A common pathological hallmark of neurodegeneration is the accumulation of cytoplasmic amyloid-like aggregated proteins [17,65,66]. Our data, utilizing two independent single nucleotide polymorphisms in *Nemf* (R86S and R487G), demonstrate that disrupted Importin-β nuclear import results in the mis-localization of nuclear proteins to the cytoplasm. Importantly, this disruption of NCT is specific to Importin-β nuclear import and not to Transportin-1 or RQC function.

Recent evidence indicates that cells actively promote the sequestration of misfolded proteins into aggregates, which are further compartmentalized within distinct cellular compartments [25,67,68]. NIRs have been shown to operate in the cytoplasm and disaggregate NLS-bearing cargoes to prevent deleterious phase transitions [69]. This ability to inhibit and reverse both physiological and deleterious phase transitions has been implicated in multiple forms of neurodegenerative diseases, wherein RNA-binding proteins or Prion-like domains are lost from the nucleus and accumulate in the cytoplasm [69–77]. This is demonstrated in the role of Importin-β cooperating with Importin-α to prevent and reverse TDP-43 condensation and fibrillization [69,71]. Furthermore, Importin-β upregulation has been shown to rescue hallmarks of TDP-43 proteinopathy [63,64]. The potential of these NIRs to serve as cytoplasmic

chaperones and restore the nuclear function of RNA-binding proteins highlights their therapeutic potential [78].

It is difficult to establish whether the mutant NEMF R86S is pathologically interacting with Importin-β directly or is indirectly associating with Importin-β through aggregate-induced sequestration. However, the absence of nuclear import defects in an *Ltn1* knockdown model highlights an RQC-independent specificity for NEMF in these observed import defects. This fact, along with the greater association of the NEMF R86S protein with Importin-β through PLA, highlights a role for NEMF interactions with Importin-β. Furthermore, the inability for RQC substrates such as GFP-Nonstop and GFP-PolyK-(AAA)$_{10}$ to properly localize to the nucleus in the *Nemf*$^{R86S}$ suggests that nuclear import mechanisms may work closely with protein degradation pathways. It has been well described that PolyK tracts are known to act as nuclear and nucleolar localization sequences [25, 79, 80], whereas the 3'UTR sequences found in the GFP-Nonstop are unable to properly localize these proteins to the nucleus and will sequester in the cytoplasm [25]. Therefore, we can infer that the inability for *Nemf*$^{R86S}$ MEFs to localize the GFP-Nonstop to the nucleus through their PolyK tracks results in their sequestration into cytoplasmic inclusions through these 3'UTR domains, whereas the GFP-PolyK and GFP-3X-NLS remain relatively diffuse. These observations prompted us to determine if NEMF's engagement in the RQC dictated the fate of these aberrant proteins. Using puromycylated nascent chains which disrupt the interaction of NEMF with the 60S ribosome [30,41], we were able to observe a similar lack of nucleolar targeting which was dependent on the ability for Importin-β to interact with its substrates. Altogether, this suggests that Importin-β's function as both a nuclear import receptor and cytoplasmic chaperone is disturbed in this *Nemf*$^{R86S}$ model, wherein the protein is sequestered away from its typical function, and that we can recapitulate many of these in vivo alterations through transient Importin-β pharmacological impairment.

The early onset of disease in the *Nemf*$^{R86S}$ mutant mice indicates that the cytoplasmic accumulation of NEMF and TDP-43 may be a result of a greater propensity to mis-localize in the R86S variant when compared to the R487G, whether through direct or indirect mechanisms. Importantly, the mis-localization of NEMF, Importin-β, Ran, RanGAP1, and TDP-43 in the mutant *Nemf*$^{R86S}$ MEFs is observed broadly throughout the cell population whereas the presence of cytoplasmic puncta is only observed in a subset of the cell population. This indicates that the nuclear import defects may precede the accumulation and sequestration of NTRs in the cytoplasm. Furthermore, the presence of insoluble TDP-43 and phospho-TDP-43 in the *Nemf*$^{R86S}$ MEFs and spinal cord motor neurons is consistent with TDP-43 pathology in neurodegeneration [81,82].

Reduced *Stmn2* expression has been implicated in human TDP-43 pathologies in ALS and AD, as well as in Parkinson's disease patients and FTLD-tau patients [62, 83–86]. Furthermore, the genetic loss of *Stmn2* alone in mice results in motor neuropathy [87, 88]. The loss of *Stmn2* transcript in our *Nemf*$^{R86S}$ mouse model suggests either that *Stmn2* is downregulated directly because of defects in nuclear import, that its down regulation is a result of the mis-localization of TDP-43 itself, or some combination of the two. The lack of a role for TDP-43 regulation of *Stmn2* in mice [83] suggests a parallel pathway in which nuclear import dysregulation can result in these differentially expressed genes. Nevertheless, our observations of pathological TDP-43 insoluble aggregates, TDP-43 missplicing events, and a downregulation of *Stmn2* mRNA and protein after a nuclear import block indicates a direct causal role of defective nuclear import upstream of these pathologies. Additional dysregulation of transcripts, such as the upregulation of *Meg3* and *Apoe*, as well as the downregulation of *Fig 4*, are seen in our model and mirror changes seen in neurodegenerative disease [51,89]. Our investigations show that by disrupting *NEMF* and *Importin-β* pathways through knockdown in human cell lines,

we can induce a significant decrease in *Stmn2* transcript. Interestingly, knockdown of *NEMF* and *Importin-α*, but not *Importin-β* causes an upregulation of *APOE* transcripts, again mimicking similar phenotypes observed in mice. The divergence for the dysregulation of *Apoe* and *Stmn2* suggests that these transcripts are more likely dysregulated through separate downstream pathways, with *Stmn2* being dysregulated by defects in nuclear import and *Apoe* by other unknown mechanisms, non-specific to neuronal cells. However, the ability for Importin-β upregulation to rescue both *Stmn2* and *Apoe* in our *Nemf*[R86S,] even if partially suggests a divergence of pathology that can be mediated by both driving nuclear import as well as the disaggregating chaperone activity of Importin-β.

We have demonstrated that there is a direct impact of Importin-β nuclear import dysregulation on multiple phenotypes seen in neurodegeneration through the following: (1) NEMF-specific dysregulation of Importin-β nuclear import (but not of Transportin-1 import), (2) mis-localization and aggregation of NEMF, TDP43, Ran, RanGAP1 and Importin-β, (3) amyloid-like aggregation, (4) PLA cytoplasmic association between NEMF R86S and Importin-β, (5) cytoplasmic accumulation of NEMF, Importin-β, and TDP43 in *Nemf* mutant mouse spinal cord, (6) transient Importin-β-specific nuclear import blockage knockdown recapitulates these transcriptional alterations (7) and upregulation of Importin-β can partially rescue these transcriptional alterations. These dysfunctions are not, however, recapitulated upon knockdown of NEMF's RQC partner LTN1, indicating a specific, non-RQC role for NEMF in impairing Importin-β nuclear import. Collectively, our data suggests that Importin-β nuclear import dysfunction serves as a primary upstream pathomechanism for TDP-43-mislocalization and phosphorylation, as well as transcriptional dysfunction seen in neurodegenerative proteinopathies.

## Methods

### Ethics statement

All mouse husbandry and procedures were reviewed and approved by the Institutional Animal Care and Use Committee at Stony Brook University (1556599) and were carried out according to the National institutes of Health Guide for Care and Use of Laboratory Animals.

### WT-NEMF and R86S-NEMF Mouse Embryonic Fibroblasts (MEFs) Derivation and Culture

Dams were euthanized by cervical dislocation. A small incision was made on the abdomen to expose the embryos. Embryos were quickly transferred to a 10cm petri dish with PBS 1X and dissected out of its respective muscular sac. The embryos were further transferred to another 10cm petri dish with PBS 1X and everything was moved to a laminar flow hood. Embryos were transferred to a fresh petri dish with Trypsin-EDTA (0.05%) and finely chopped into fine pieces with a fresh blade and placed at 37˚C, 5% $CO_2$ for 5–10 minutes. Chopped embryos were passed through a 20 ½ gauge syringe. Fresh Media (Dulbecco's Modified Eagle's Medium (10% FBS, Glutamine 1X (Glutamax), PenStrep)) was used to neutralize the trypsin and cells were transferred into a 50mL tube where the cells were triturated 5–10 times before being divided and placed into a dish and placed back into a 37˚C, 5% CO2 incubator for 3 days. Media was changed once on day 3 and cells were passaged for 14 days until they were frozen back in liquid nitrogen. WT NEMF and R86S-NEMF MEFs were maintained in Dulbecco's Modified Eagle's Medium (10% FBS, Glutamine 1X (Glutamax), PenStrep) at 37˚C, 5% $CO_2$ and passaged every two days with Trypsin-EDTA (0.05%). Reagents used indicated in S1 Table.

## HEK293 Culture

HEK293 cells were maintained in Dulbecco's Modified Eagle's Medium (10% FBS, 1mM Glutamax, PenStrep, Non-essential amino acids) at 37˚C, 5% $CO_2$ and passaged every two-four days with Trypsin-EDTA (0.05%). Reagents used indicated in S1 Table.

## SK-N-MC Culture

SK-N-MC cells (ATCC) were cultured according to the vendor's protocol. In brief, cells were maintained in ATCC-formulated Eagle's Minimum Essential Medium (10% FBS, PenStrep) at 37˚C in 5% $CO_2$. Reagents used indicated in S1 Table.

## Microplate MEF Plating

Coverslips (Carolina Biological Supply) were washed with 70% ethanol in a sterile petri dish to sterilize and then allowed to air dry for 30mins. Using a 24 well microplate, the sterilized coverslips were then placed inside each well, and then 1mL of the 0.1%(w/v) gelatin (Sigma) was added and incubated at 37˚C for 1hr. The excess gelatin was aspirated with a vacuum and the coverslips were again incubated at 37˚C for 1hr.

## Plasmid and SiRNA Transfections

Transfection of plasmids and siRNAs (S2 Table) was performed using Lipofectamine 3000 and Lipofectamine 2000 (Life Technologies) following manufacturer's instructions. Experiments were performed 48 hours after transfections of plasmids and siRNAs. Control transfections were performed using a scrambled control.

## Nuclear Import Block

Cells were treated with 20uM and 10uM of importazole (Sigma) and ivermectin (Sigma) respectively for 48 hours at 37˚C 5% $CO_2$.

## Immunofluorescent Staining

Cells were chemically fixed in 4% paraformaldehyde in PBS(1X). Each well was then rinsed with PBS(1X). The cells were then permeabilized with 0.1% TritonX-100/PBS for 10mins at room temperature. The cells were then blocked at room temperature with 5% NGS/PBS/0.1% Tween-20. The cells were then incubated with the respective primary antibody in 5% NGS/0.1% Tween-20/PBS overnight at 4˚C. The primary antibodies used are listed in S3 Table. The next day, the cells were rinsed 3 times for 5mins with 0.1% Tween-20/PBS(1X). The cells are then incubated with the respective secondary antibodies in 5% NGS/0.1% Tween-20/PBS for 2hrs in the dark at room temperature. The secondary antibodies used are listed in S4 Table. The cells were then rinsed 3 times for 5mins with 0.1% Tween-20/PBS and stored temporarily with 200µL of PBS(1X) in the dark at 4˚C. Using sterile slides, 20–25µL of Prolong glass with NucBlue (Thermo Fisher) was added to the slide. The liquid was then aspirated from the well and using an SE 5 tissue curved forceps, the coverslips were gently picked up and then placed with the cell layer (top) down on the slide. The slide is then cured in the dark at 4˚C for 24hrs. Confocal images were taken by an Olympus FV1000 Laser Scanning Confocal. Laser settings (laser strength, gain, and offset) and magnification were maintained across treatment groups. Post-processing of images was performed by ImageJ and Cell Profiler as described below.

## Proximity Ligation Assay (PLA)

PLA was performed as described in the DuoLink In Situ PLA protocol (Millipore Sigma). In brief, cells were seeded onto gelatin coated coverslips and grown overnight. Cells were chemically fixed in 4% paraformaldehyde in PBS (1X). Each well was then rinsed with PBS (1X). The cells were then permeabilized with 0.1% TritonX-100/PBS for 10mins at room temperature. Cells were blocked for 60 minutes in DuoLink Blocking Solution at 37˚C for 60mins. Cells were then incubated overnight at 4˚C with the respective primary antibody combination in Duolink Antibody Diluent. Cells were washed twice in Duolink Wash Buffer A and then incubated in PLA probe solution for 60mins at 37˚C. Cells were then washed twice in Duolink Wash Buffer A and then incubated in ligation solution for 30mins at 37˚C. Cells were then washed twice in Duolink Wash Buffer A and then incubated in amplification solution for 100mins at 37˚C. Cells were then washed twice in Duolink Wash Buffer B and washed once in 0.01X Wash Buffer B for 1 minute. Cells were then Mounted with Duolink PLA mounting media with DAPI. Confocal images were taken by an Olympus FV1000 Laser Scanning Confocal. Laser settings (laser strength, gain, and offset) and magnification were maintained across treatment groups. Post-processing of images was performed by ImageJ and Cell Profiler as described below.

## Dextran nuclear exclusion analysis

Analysis of the Nuclear Pore Complex using the plasma membrane permeabilization was performed as previously described [90]. In brief, cells were seeded onto gelatin coated coverslips and grown overnight. Optimization of permeabilization was performed first on 3 test coverslips. Cells were washed once in PBS (1X) at 37˚C. Microplates were placed on ice and cells were washed once in ice cold PBS (1X). Permeabilization buffer (20 mM HEPES pH 7.5, 110 mM KOAc, 5 mM $MgCl_2$, 0.25 M sucrose, and protease inhibitors) was then added to the cells and incubated for 5mins. The buffer was then replaced with permeabilization buffer containing 20ug/mL of digitonin and incubated for 4, 7, and 10mins for each coverslip to optimize permeabilization time. Cells were then washed 4 times with diffusion buffer (20 mM HEPES pH 7.5, 110 mM KOAc, 5 mM sodium chloride, 2 mM MgCl2, 0.25 M sucrose, and protease inhibitors). 2μl of diffusion assay solution (diffusion buffer with 0.6mg/mL 500kDa FITC-dextrans) was placed onto a glass slide and coverslips were lifted and mounted onto the slide. The coverslips were sealed with clear nail polished, incubated for 15mins and imaged on the Olympus FV1000 Laser Scanning Confocal. These steps were repeated with 60-70kDa and 500kDa FITC-dextrans at the optimal permeabilization time where the 500kDa FITC-dextrans were found in cytoplasm but excluded from the nucleus. Laser settings (laser strength, gain, and offset) and magnification were maintained across treatment groups. Post-processing of images was performed by ImageJ and Cell Profiler as described below.

## Puromycylation of Nascent Chain Polypeptides

Cells were grown on gelatin-coated glass coverslips as described above. Labeling of newly synthesized proteins was performed by incubating the cells in 25uM O-propargyl-puromycin (OP-Puro) (BioVision) for 2hrs with or without Importazole (40μM) (Sigma). Samples were incubated for 2hrs with each treatment. Following the incubation period, cells were rinsed once with PBS(1X) and then chemically fixed in 4% PFA in PBS(1X) for 10mins at room temperature and permeabilized for 15mins with 0.1% TritonX-100 in PBS(1X). Labeling of the OP-Puro-labeled peptides for immunolabeling was performed following EZ Click Global Protein Synthesis Assay Kit Protocol. In brief, cells were incubated for 30mins at room temperature in EZClick Reaction cocktail and then washed for once in PBS. Immunostaining occurred

after 'Click' Chemistry reaction. Confocal images were taken by an Olympus FV1000 Laser Scanning Confocal. Laser settings (laser strength, gain, and offset) and magnification were maintained across treatment groups. Post-processing of images was performed by ImageJ and Cell Profiler as described below.

### Growth assay

Cells were seeded on a 24 well plate at a cell density of 10,000 cells per well. Using Agilent Lionheart Imager, three regions of interests in the well were imaged for 48 hours every 15mins. Cell growth was standardized by starting cell number.

## RIPA Lysis Protein Extractions

Cells were washed twice with 5mL of pre-chilled PBS(1X). 500μL of RIPA lysis buffer (Sigma) supplemented with protease inhibitor was added to the flask and then incubated on ice for 5mins. The cells were scraped and then transferred to a 1.5mL microtube. The lysate was then agitated for 30mins at 4˚C and then centrifuged for 20mins at 12,000rpms at 4˚C. The supernatant was then transferred to a pre-chilled microtube and stored at -80˚C.

## Sarkosyl Insoluble Fractionation

Pathological TDP-43 aggregates were biochemically isolated as previously described in Manuela et al 2019 [50]. In brief, cells grown in a 6 well plate were washed twice in 1ml of PBS (1X). 100ul of 1X HS Buffer (10mM Tris, pH 7.4, 15mM NaCl, 0.5mM EDTA, 1mM DTT, protease and phosphatase inhibitors) supplemented with 0.5% Sarkosyl (Sigma) and Benzonase Endonuclease (12mM $MgCl_2$ with 250U/Sample Benzonase) (Sigma) was added to the wells and the cells were scraped into a 1.5ml microtube. The wells were washed 1X with 1X HS Buffer, 0.5% Sarkosyl and added to the microtube. 200ul of 2X HS Buffer, 4% Sarkossyl was added to the lysate and incubated on ice for 45mins with vortexing in 10min intervals. Following cell lysis, 200ul of ice cold 1X HS Buffer, 0.5% Sarkosyl was added the sample and then centrifuged at max speed for 20mins at RT. The supernatant was collected as the soluble fraction, and the pellet was washed twice with PBS 1X. Pellets were frozen at -80˚C until SDS PAGE.

### Pierce BCA protein assay

The standards were prepared according to the Test Tube protocol provided by the Pierce BCA (Thermo Fisher). 10μL of each standard was pipetted into appropriately labeled 500μL microtubes and repeated for each label. 200μL of working reagent was added to each tube and incubated in a water bath at 37˚C for 30mins. The standards were analyzed on the NanoDrop at 562nm to establish a standard curve and the samples were then analyzed on the curve.

### SDS-PAGE

10μg of lysate was prepared with 50mM dTT (Biorad) and 4X laemli buffer (Biorad). The solution was vortexed quickly and boiled for 5mins at 95degC. The lysates were then chilled briefly on ice and then loaded onto a Stain-Free mini-protean 10 well pre-cast gel (BioRad) and mini-protean tank (BioRad) with a Chameleon 800 MW ladder (Licor). The gels were run at 200V for approximately 45mins in Tris/Glycine/SDS Running Buffer (BioRad). Gels were then equilibrated in transfer buffer (20% methanol) (BioRad) and total protein was imaged using a ChemiDoc Imaging System (BioRad).

## Semi-dry Transfer (PVDF .2um)

The PVDF membrane was pre-wet in methanol, then rinsed with ultra-pure water. The membrane and 6 filter papers are equilibrated for 20mins in transfer buffer. A gel sandwich is then prepared with 3 filter papers, gel, membrane, 3 filter papers, respectively. The transfer is then run for 90mins, with the current monitored to maintain 80mA-240mA. After the transfer, the membrane is removed and allowed to dry completely. The membrane is then re-activated with methanol, and rinsed once with TBS (1X), then stored in TBS(1X) in the fridge until immunoblotting.

## Immunoblotting

The membrane is rinsed with TBS(1X) and then incubated with TBS-Based Odyssey blocking buffer for 1hr at room temperature with gentle rocking. The blocking buffer is then discarded, and the membrane is incubated in blocking buffer supplemented with 0.1% Tween-20 and primary antibody overnight with gentle rocking. The primary antibodies used are listed in the S3 Table. The membrane is then washed with TBS-T (0.1% Tween-20) 3 times for 5mins with gentle shaking. The membrane is then incubated with blocking buffer (0.1% Tween-20) and the respective secondary antibody at room temperature for 2hrs with gentle rocking. The secondary antibodies used are listed in the S4 Table. The membrane is then washed with TBS-T 3 times for 5mins. The membrane is then rinsed with TBS(1X) and imaged with the Odyssey Scanner (Licor).

## Mouse strains, husbandry, and genotyping

Tail tissue was lysed in proteinase K at 55deg C overnight and extracted DNA was used for genotyping. Genotyping for B6J-$Nemf^{R86S}$ was performed via PCR using the following primers: forward primer specific to wild-type allele: 5′-AACATTTGAAGAGTCGGGGA-3′; forward primer specific to mutant allele: 5′-AACATTTGAAGAGTCGGGGT-3′; reverse primer common for both alleles: 5′-GCAGGTGGATGGTAGCAACG-3′. Similarly, for the genotyping of the B6J-$Nemf^{R487G}$ mice the following primers were used: forward primer specific to wild-type allele: 5′-TGCTGCTAAAAAAACCCGGA-3′; forward primer specific to mutant allele: 5′-TGCTGCTAAAAAAACCCGGG-3′; reverse primer common for both alleles: 5′-AA AGCCCTTGCTGCAAAGCC-3′.

## Primary neuron isolation and culture

P0/P1 pups were euthanized by cervical dislocation. Cortices were quickly dissected from the brain in 2mL of Hibernate-A/B27(0.5mM GlutaMAX, PenStrep, 1% B27) at 4˚C in a 35mm dish. Cortices were minced with as scalpel and transferred into a tube containing Papain Digestion Medium (100units Papain in 2mL Hibernate-A) and incubated for 30mins at 30˚C and agitated on a platform shaker at 170rpm. Slices are transferred to a 15mL conical containing 2mL of Hibernate-A/B27 and allowed to settle for 5mins. Slices are titurated 10 times with a siliconized 9 inch Pasteur pipette with a tip fire polished to an opening of 0.7 to 0.9μm diameter. Slices are allowed to settle for the supernatant to be transferred to a new tube, and the tituration is repeated in another 2mL of Hibernate-A/B27. The cells in the supernatant were pelleted by centrifugation at 80 x g for 5mins. The remaining supernatant was discarded, the cells were gently resuspended in 1-3mL of Hibernate-A/B27. Cells were then counted by hemocytometer. The cells were further pelleted by centrifugation at 80 x g for 5mins. The supernatant was discarded, and the cell pellet was resuspended in an appropriate volume of Neurobasal-A/B27(B27 1X, PenStrep, 1X GlutaMax) and the cells were plated at 80% of plating

volume. The medium was removed 30-45mins post plating and replaced with fresh medium to prevent debris from settling along with the cells. Cells were maintained at 37°C in 5% $CO_2$ and cultured for 2 weeks with media changes every 2 days.

## Spinal Cord Isolation

Mice were euthanized by carbon dioxide inhalation and immediately exsanguinated by cardiac puncture using a 1mL syringe with a 25 gauge needle. Using small surgical scissors and forceps, the dorsal skin was removed. Next, the location of the atlanto-occipital joint was visualized by repeated flexion and extension of the spinal column and palpation. Using large scissors, animals were decapitated at the atlanto-occipital joint, exposing the cervical spinal cord. Following decapitation, a clean transverse cut was made through the lumbar portion of the spine, cranial to the iliac crest. Moving from caudal to cranial, the vertebral column was dissected from the ventral portion. The entire vertebral column was rinsed with PBS 1X and placed in a 15mL conical tube with 4% PFA for 20mins at 4°C. During this stage, skull cap is removed with brain gently teased from the cranial cavity in a rostro caudal direction. Micro-dissection forceps are used to gently tease apart the cranial nerves. The brain is then rinsed in PBS 1X and placed in a 15ml conical tube with 4% PFA at 4°Cwith agitation overnight. The vertebral column stored in 4% PFA is then placed into a 100mm petri dish with PBS 1X. Moving from caudal to cranial, fine-tipped offset bone nippers were used to reveal the dorsal aspect of the spinal cord. The vertebral column is then held vertically, and the spinal cord is eased from the spinal canal using micro-dissection forceps and placed into the 100mm petri dish. The spinal cord was then placed into a new 15ml conical tube with 4% PFA at 4°C with agitation overnight. The following day, the brain and spinal cord are both transferred to 30% sucrose (w/v) for an additional 24h. The following day, the spinal cord is sectioned above the lumbar enlargement region with microscissors. The brain, the lumbar region and the thoracic and cervical regions are both placed in optimal cutting temperature (OCT) and stored in -80°C.

## Cryosectioning

Lumbar spinal cords and cervical and thoracic spinal cords stored in OCT were placed in -20°C for one hour prior to cryosectioning to allow for the temperature to equilibrate. Spinal cords were then sectioned transversely in the cranial to caudal direction at 30μm slices and placed free-floating in a 24 well microplate filled with PBS 1X at 20 sections per well. Sections were allowed to equilibrate in PBS 1X for 1 hour. Sections were then rinsed with PBS 1X to remove residual OCT and were then redistributed to a 48 well plate in PBS for immunoblotting.

## Spinal cord immunoblotting

Spinal Cords were rinsed with PBS (1X). Spinal Cords were then permeabilized in 0.3% TritonX-100/PBS 1 hour at room temperature with agitation. The spinal cords were then blocked at room temperature with 5% NGS/PBS/0.1% TritonX-100. The spinal cords were then incubated with the respective primary antibody in 5% NGS/PBS/0.1% TritonX-100 overnight at 4degC. The primary antibodies used are listed in S4 Table. The next day, the spinal cords were washed 3 times for 15mins with 0.1% TritonX-100/PBS(1X). The spinal cords are then incubated with the respective secondary antibodies in 5% NGS/PBS/0.1% TritonX-100 for 2hrs in the dark at room temperature. The secondary antibodies used are listed in S4 Table. The spinal cords were then washed 3 times for 15mins with 0.1% TritonX-100/PBS(1X). The spinal cords were rinsed with PBS 1X. The spinal cords were then rinsed with 70% ethanol, and then

incubated with Auto-fluorescent eliminator (EMD Millipore) for 5mins. The spinal cords were then rinsed 2 times for 5mins with 70% ethanol. Spinal cords were rehydrated in PBS (1X) for 5mins and then transferred to phosphate buffer (1X). Using a dissecting microscope, spinal cord sections were then mounted onto Superfrost Plus charged slides (Fisherbrand) and allowed to dry. 200μL of Fluoromount with DAPI (Sigma) was then added to the slide and a #1.5 22x48mm coverslip (Fisherbrand) was mounted on top. Confocal images were taken by an Olympus FV1000 Laser Scanning Confocal. Laser settings (laser strength, gain, and offset) and magnification were maintained across treatment groups. Post-processing of images was performed by ImageJ and Cell Profiler as described below.

## RNA extraction

Spinal cord and brain tissue was isolated as described above and then frozen at -80˚C. Tissue samples were homogenized in 1ml of TRIZOL reagent with a power homogenizer (Fisher Scientific). The sample was then transferred to a phase lock gel tube, shaken vigorously, and incubated at room temperature for 5mins. 200μL of chloroform was added, shaken vigorously again for 15s and then incubated for 2mins at room temperature. The sample is then centrifuged for 15mins at 12,000gs at 4˚C. The aqueous phase is then mixed with an equal volume of ethanol and then the RNA is isolated using a PureLink RNA mini kit by manufacturer's instructions.

## qPCR

RNA (200ng) was reverse transcribed (Superscript IV Reverse Transcriptase (Thermo Fisher)) and the output volume of 20μL was diluted in nuclease-free water to 40μL for a working concentration of 5ng/μL. Real time PCR was performed using SYBR Green PCR Master Mix (Applied Biosystems) on a QuantStudio 3 System (Applied Biosystems) with reaction specificity confirmed by melt curve analysis. All comparisons (Control vs Experimental) for each qPCR reaction was run on the same qPCR plate and was run in a triplicate. For qPCR primer sequence, see S6 Table.

## Tissue protein extraction

Spinal cord and brain tissue was isolated as described above and then frozen at-80˚C until protein extraction. Tissue samples were homogenized with a microtube pestle in 500ul of RIPA lysis buffer (Sigma) supplemented with protease inhibitors. The tissue was then passed through a 70μm cell filter and collected in a 1.5mL microtube. The lysate was then agitated at 300rpms for 30mins at 4˚C and then centrifuged for 20mins at 12,000rpms at 4˚C. The supernatant was then transferred to a pre-chilled microtube and stored at -80˚C.

## RNA Seq of NEMF$^{R86S}$ MEFs

Concentrated RNA was sent for bulk RNAseq to Azenta. In brief, sample quality control and determination of concentration was performed using TapeStation Analysis by Azenta, followed by library preparation and sequencing. Computational analysis included in their standard data analysis package was used for data interpretation.

## Image analyses

Images were analyzed in bulk through Cell profiler. Z-projections were taken from each image by maximum intensity and then separated by fluorophore. Nuclear/Cytoplasmic (N/C) Ratios

were taken by comparing area of nucleus (DAPI) and area of the cytoplasm (Phalloidin or brightfield). Nuclear and Cytoplasmic Intensities were standardized to area.

## Supporting information

**S1 Fig. *Nemf*^R86S^ Mice display nuclear loss of NEMF in the primary motor cortex and show a specific motor neuron defect in the spinal cord.** A) Primary Motor Cortex was isolated from 21-day old Wild Type and *Nemf*^R86S^ mice. Neurons in the ventral horn were immunostained for the nucleus (DAPI, blue) and NEMF (red). B) Nuclear/Cytoplasmic ratios of NEMF in WT and Nemf^R86S^. Data analyzed by unpaired two-tailed t-test. C-D) Lumbar Spinal Cords were immunostained for the nucleus (DAPI, blue) and neurons (Nissl, red), and NEMF (C) and TDP-43 (D) (green). Arrows indicate pathological NEMF or TDP-43 Nissl-positive neurons. Individual colors in plots represent one trial. (**** p<0.0001).
(TIF)

**S2 Fig. *Nemf*^R86S^ MEFs display Importin-β specific nuclear import defects and a 'leaky' nuclear pore.** A) Western Blot Analysis of GFP and GFP-3X-NLS expression in WT and *Nemf*^R86S^ MEFs, and in LTN1 and NEMF siRNA treated WT MEFs. B) Expression of GFP-3X-NLS co-stained with Importin-β in WT MEFs. C-D) qPCR validation of *Ltn1* and *Nemf* siRNA knockdown (n = 3). E) Normalized Cell survival of WT and *Nemf*^R86S^ MEFs expressing *LTN1* and *NEMF* siRNA (n = 3). F) Expression of GFP, GFP-3X-NLS, and mCherry-PY-NLS co-stained with Importin-β in WT MEFs treated with IPZ. G) FITC-conjugated dextrans (60-70kDa, and 500kDa). Nuclei labeled with DAPI (blue). H) Quantification of the integrated intensity of dextrans within the nucleus. Individual colors in plots represent one trial. (ns p>0.05, ** p<0.01).
(TIF)

**S3 Fig. Negative Controls for NEMF and Importin-β PLA** A) PLA of NEMF and Importin-β alone, or without primary antibodies (red). Nuclei labeled with DAPI (blue).
(TIF)

**S4 Fig. *Nemf*^R86S^ MEFs show cytoplasmic mis-localization nuclear transport factors and an increase in an amyloid-like phenotype.** A) Quantification of WT and *Nemf*^R86S^ MEF cell size in square microns using phalloidin as cell maker. Data analyzed by unpaired two-tailed t-test. (n = 999–1000) B) Growth curves of WT (n = 3, $r^2$ = 0.99) and Nemf^R86S^ (n = 3, $r^2$ = 0.98) over 48 hours at 15min intervals. Data was analyzed by a nonlinear regression for Malthusian growth. C) Western Blot analysis of global protein synthesis by puromycin immunostaining with or without cycloheximide (CHX) pre-treatment. D) Quantification of puromycin immunostaining standardized to stain-free total protein stain (n = 3). E) Quantification of the percentage of cells with cytoplasmic puncta from Fig 4. Data was analyzed by two-way ANOVA with Šídák's multiple comparisons test (n = 3). F-J) Quantification of Nuclear/Cytoplasmic Ratio of indicated proteins. Data analyzed by unpaired two-tailed t-test (n = 100). K) Staining of amyloid in WT and *Nemf*^R86S^ MEFs by AmyloGlo. Nuclei labeled with Lamin A/C. L) Quantification of the Integrated Intensity of AmyloGlo. Data analyzed by two-tailed t-test. (n = 30–60) M) Normalized Cell Survival of IPZ treated WT and *Nemf*^R86S^ MEFs (n = 3). Individual colors in plots represent one trial. (ns p>0.05, **p<0.01, ***p<0.0001, ****p<0.0001).
(TIF)

**S5 Fig. *Nemf*^R86S^ MEFs show cytoplasmic mis-localization of nucleoporins, but not Exportin-1 or Transportin-1.** A) Immunostaining of Exportin-1 or Transportin-1 (red) in WT and *Nemf*^R86S^ MEFs. Nuclei labeled with DAPI (blue). B) Immunostaining of NEMF (green) and

NPM1 (red) in WT and *Nemf*^R86S MEFs. C) Immunostaining of RPL3 (60S, green) and RPS6 (40S, red) in WT and *Nemf*^R86S MEFs. D) Immunostaining of Nup153, and Nup98, and mAb414(green) co-stained with Importin-β (red). Nuclei labeled with DAPI (blue). E) Percentage of Nuclear Signal of Nup153, Nup98, and mAb414 in WT and *Nemf*^R86S MEFs (n = 30–33 nuclei). Individual colors in plots represent one trial. Scale bars are 20μm. (TIF)

**S6 Fig.** Late onset *Nemf*^R487G spinal motor neurons show appearance of phospho-TDP-43 at 6- and 12-months A) Immunofluorescent staining of phospho-TDP-43 (red), ChAT (green) in WT, and NemfR487G lumbar spinal cord motor neurons at 21 days, 6 months, and 12 months. Nuclei labeled with DAPI (blue). B) Percentage of ChAT+ cells with pTDP-43 cytoplasmic inclusions in WT and NemfR487G mice (n = 3). Data analyzed by two-way anova with Šídák's multiple comparisons test. (n = 3). (ns p>0.05, *** p<0.001). (TIF)

**S7 Fig.** A-B) qPCR relative fold change of *Bax* and *Sorl1* in MEFs, spinal cord, brain in WT and *Nemf*^R86S mice. Data analyzed by two-way anova with Šídák's multiple comparisons test. (n = 3). C) Relative expression of Sort1 and Dnajc5 exon splicing inclusion from WT and *Nemf*^R86S MEFs cDNA. D) Brightfield Images of WT and *Nemf*^R86S Primary Neurons at 14 days post-plating. E-F) qPCR relative fold change of *Stmn2* and *Apoe* in WT and *Nemf*^R86S primary neurons (n = 3). G) Gene Ontology Analysis of significantly dysregulated genes in Nemf^R86S (ns p>0.05, *p<0.05, **p<0.01). (TIF)

**S8 Fig. RNA Quality is maintained high through all reads.** A) qPCR log2FoldChange plotted over RNA for MEFs ($r^2$ = .76,*p<0.05), Spinal Cord (SC) ($r^2$ = .89,**p<0.01), and Brain ($r^2$ = .05,*p>0.05). A simple linear regression determined the best-fit model. B) Tapestation Analysis comparing RNA integrity number for WT, IPZ-treated, and IVM-treated RNA samples. Data analyzed by one-way anova with Tukey's multiple comparison test. C) RNA quality scores throughout Sanger Sequencing Position as presented by Azenta. (ns p>0.05). (TIF)

**S9 Fig. IPZ and IVM Cell Viability Curves and Gene Ontology Analyses** A-B) Cell Viability Curves for IPZ (A) and IVM (B) treated MEFs. Data analyzed by one-way with Tukey's multiple comparison test C) Percentage of cells with pTDP-43 cytoplasmic inclusions in DMSO-CTRL and IPZ-treated WT MEFs (n = 3). E) Gene Ontology Analysis of significantly dysregulated genes in Nemf^R86S and IPZ-treated samples. F) Brightfield Images of DMSO-CTRL and IPZ-treated WT Primary Neurons at 14 days post-plating. G-H) qPCR relative fold change of *Stmn2* and *Apoe* in WT and IPZ-treated primary neurons (n = 3). (ns p>0.05, *p<0.05, **p<0.01, **** p<0.0001). (TIF)

**S10 Fig. qPCR Validation and *STMN2* expression in HEK293 cells** A) Immunofluorescent staining of TDP-43 and pTDP-43 in DMSO control and Importazole-treated (IPZ) SK-N-MC (neuroblastoma) cells. B) Quantification of Nuclear/Cytoplasmic Ratio of TDP-43 (n = 80–96). C) Percentage of cells with pTDP-43 cytoplasmic inclusions in DMSO contorl and IPZ-treated (n = 3). D) qPCR relative fold change in DMSO control and IPZ-treated SK-N-MC cells for *STMN2* (n = 3). E) qPCR validation of *NEMF*, *LTN1*, and *Importin-β* siRNA knockdown in SK-N-MC cells (n = 3). F-G) qPCR relative fold change of *STMN2* or *APOE* treated with *LTN1*, *NEMF*, or *Importin-β* siRNAs in SK-N-MC cells (n = 3). Data analyzed by one-way ANOVA with Tukey's multiple comparison test. H) qPCR relative fold change in DMSO

Control and Importazole (IPZ) treated HEK293 for *STMN2* (n = 3). I) qPCR validation of *LTN1 (RNF160)*, *NEMF* (*SDCCAG1*), *Importin-β* (*KPNB*), Importin-α (KPNA), or *Transportin-1* (TPNO1) siRNA knockdown in HEK293 cells (n = 3). Data analyzed by two-way ANOVA with Šídák's multiple comparisons test. J-K) qPCR relative fold change of *STMN2* or *APOE* treated with *LTN1*, *NEMF*, *Importin-β*, Importin-α, or *Transportin-1*siRNAs in HEK293 cells (n = 3). Data analyzed by one-way anova with Tukey's multiple comparison test. Individual colors in plots represent one trial. (ns p>0.05, *p<0.05, **p<0.01, ***p<0.001 **** p<0.0001).
(TIF)

**S1 Table. List of Cell Culture Reagents.**
(TIF)

**S2 Table. List of Plasmid and SiRNAs.** GFP-Stop, GFP-Nonstop, and GFP-PolyK-(AAA)$_{10}$ plasmids were previously described in Davis *et al*, 2021 [25].
(TIF)

**S3 Table. List of Primary Antibodies.**
(TIF)

**S4 Table. List of Secondary Antibodies.**
(TIF)

**S5 Table. List of Products used in Experiments.**
(TIF)

**S6 Table. List of Primers.**
(TIF)

**S1 Data. Numerical Data Files.**
(ZIP)

## Acknowledgments

We wish to thank current and former members of the Sher lab for critical discussions and advice. We wish to thank Dr. Gregory Cox (Jackson Laboratory and University of Maine) for help with providing NEMF mutant mice and with suggestions for research strategies. We thank Dr. Joshua Dubnau for experimental design advice. We thank Dr. Onn Brandman for the kind gift of the GFP stalling reporters. We thank Dr. Leonard Petrucelli (Mayo Clinic) for the kind gift of the mouse-reactive phospho-TDP43 antibody. We thank Wendy Ackmentin for her incredible help on all aspects of keeping the lab facilities running.

## Author Contributions

**Conceptualization:** Jonathan Plessis-Belair, Roger B. Sher.

**Data curation:** Jonathan Plessis-Belair.

**Formal analysis:** Jonathan Plessis-Belair, Roger B. Sher.

**Funding acquisition:** Roger B. Sher.

**Investigation:** Jonathan Plessis-Belair, Kathryn Ravano, Ellen Han, Aubrey Janniello, Catalina Molina, Roger B. Sher.

**Methodology:** Jonathan Plessis-Belair, Kathryn Ravano, Ellen Han, Aubrey Janniello, Catalina Molina, Roger B. Sher.

**Project administration:** Roger B. Sher.

**Resources:** Roger B. Sher.

**Validation:** Roger B. Sher.

**Visualization:** Jonathan Plessis-Belair, Roger B. Sher.

**Writing – original draft:** Jonathan Plessis-Belair, Roger B. Sher.

**Writing – review & editing:** Jonathan Plessis-Belair, Roger B. Sher.

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
