## [Decision Letter · Decision Letter 0]

20 May 2024

Dear Dr Sher,

Thank you very much for submitting your Research Article entitled 'Importin-β specific nuclear transport defects recapitulate phenotypic and transcriptional alterations seen in neurodegeneration' to PLOS Genetics.

The manuscript was fully evaluated at the editorial level and by independent peer reviewers. The reviewers appreciated the attention to an important problem, but raised some substantial concerns about the current manuscript. Based on the reviews, we will not be able to accept this version of the manuscript, but we would be willing to review a much-revised version. We cannot, of course, promise publication at that time.

If you decide to revise the manuscript for further consideration at PLOS Genetics, please aim to resubmit within the next 60 days, unless it will take extra time to address the concerns of the reviewers, in which case we would appreciate an expected resubmission date by email to plosgenetics@plos.org.

We are sorry that we cannot be more positive about your manuscript at this stage. Please do not hesitate to contact us if you have any concerns or questions.

Yours sincerely,

Gregory A. Cox

Academic Editor

PLOS Genetics

Scott Williams

Section Editor

PLOS Genetics

Reviewer's Responses to Questions

**Comments to the Authors:**

Reviewer #1: see attached review

Reviewer #2: Altered nucleocytoplasmic transport (NCT) capacity has been implicated in multiple neurodegenerative diseases. However, it remains unclear as to whether this phenomenon is a driver of toxicity in disease. In this study, Plessis-Belair and colleagues identify and characterize NCt alterations and hallmarks of neurodegenerative pathology in a mouse model based on the expression of mutant NEMF, a gene previously linked to neurodevelopmental alterations. The authors demonstrate that this model shares many features of neurodegenerative disease including impaired nuclear import, TDP-43 pathology, and transcriptional alterations. Additionally, the authors provide evidence that transient inhibition of importin B alone is sufficient to trigger disease associated pathology. While of potential interest and significance to a broad readership, there are a number of improvements that should be made to strengthen the overall conclusions and completeness of the study prior to publication. These are detailed below.

1. The study and/or manuscript would benefit from a better description of the mouse model used. For example, is mutant NEMF expressed throughout development? Is this a conditional model? Is mutant NEMF expressed in all cell types? Is mutant NEMF OE or is this a CRISPR generated mutation? Can the authors please detail additional hallmarks of neurodegeneration such as neuronal loss, GFAP immunoreactivity, Iba1 immunoreactivity, survival deficits, behavioral abnormalities etc?

2. Are the phenotypes observed in MEFs recapitulated in CNS cell types (ie primary neuron or glial cultures from the mutant mice)?

3. Cell based studies appear rely on OE of mutant NEMF. Could the authors provide evidence that endogenous levels of mutant NEMF induce similar pathophysiological phenomena?

4. The authors state that Nups and NTRs are “sequestered” in cytoplasmic puncta in the context of mutant NEMF expression. However, there is little evidence to support this. In particular, the typical expression patterns of the proteins evaluated appear to remain largely intact. Thus, it is unclear if this is truly sequestration away from normal function or simply a few molecules trapped in abnormal cytoplasmic puncta. What percentage of molecules are in puncta as opposed to normal function and is this sufficient to recapitulate inhibition or loss?

5. Does OE or activation of importin B reverse or prevent pathologic alterations in mutant NEMF model?

6. Although NCTs reporters suggest that transportin mediated nuclear import and nuclear export are not disrupted, can the authors please perform immunostaining to determine if their localization is intact or altered?

7. For the dextran assay, it is unclear why the authors decided to isolate nuclei as opposed to using more conventional permeabilized cell assays. Importantly, the nuclei look rather disrupted and dextran molecules in and outside of the nucleus are hard to see given the lack of individual channel images presented.

8. Are mutant NEMF cells dying in culture? Many images presented appear to contain a number of unhealthy looking cells.

9. IPZ and IVM are clearly toxic to cells. Can the authors please comment on whether transcriptomic changes are truly a reflection of the effects of impaired NCT or simply a reflection of cell death cascades?

10. Can the authors please demonstrate that their importazole treatment blocks import (importin B vs transportin) as expected via use of NCT reporters?

11. siNEMF recapitulates transcriptional alterations but earlier experiments presented suggest that NEMF KD does not recapitulate mutant NEMF phenotypes. These seemingly discrepant results would benefit from additional discussion.

12. NEMF and Importin B staining and localization appear quite different in MEFs (Fig 3) vs in vivo (Fig 1). Staining in motor cortex (S1) vs spinal cord (Fig 1) also appear quite different even in WT mice. Can the authors comment on this and/or the specificity of antibodies used?

13. PLA results are difficult to interpret as presented. It appears the majority of the NEMF signal is within the nucleus and importin B granules are largely not colocalizing with cytoplasmic NEMF. Yet there are abundant PLA signals in the cytoplasm. Can the authors please comment on antibody specificity and include higher magnification images of no primary and no probe controls for PLA to ensure specificity of all signals?

14. In S5, much of the pTDP signal appears to fill the nucleus in an age dependent manner. This is not typical of pTDP immunoreactivity in other neurodegenerative mouse models where signal is observed as cytoplasmic puncta. Can the authors please comment on this and/or verify their findings with conventional DAB based IHC?

15. TDP-43 regulatory sequences are not conserved in mouse vs human STMN2 RNA. Thus, it is well known that STMN2 does not represent a TDP-43 target in mouse models. As a result, it is unclear whether downregulation of STMN2 in NEMF MEFs actually reflects TDP-43 loss of function pathology. It is suggested that the authors evaluate bonafide mouse TDP-43 targets.

16. Could the authors please speculate on the mechanisms by which mutant NEMF or WT NEMF may contribute to altered NCT and neurodegeneration in the discussion?

17. Overall, the title and abstract are written in a manner that suggests the focus of this manuscript is on defining the role of importin B in neurogenerative disease. However, the data and manuscript are largely focused on characterizing NCT defects in an NEMF mouse model. Thus, it is suggested that the title and abstract be re-written to more accurately represent the data presented in this manuscript.

Reviewer #3: In this paper, Plessis-Belair et al. collect evidence that Importin-beta transport defects elicit phenotypic and transcriptional alterations connected with TDP-43 proteinopathies. This study is high quality and quite compelling. Overall, it helps build a narrative that Importin-beta nuclear import dysfunction is likely to be a pathological contributor to disease. I recommend publication once a couple of key issues are addressed:

1. It remains possible that the Importin-β transport defects are a consequence rather than a cause of disease in the NEMF mouse model. Hence It is important to demonstrate that correcting the Importin-beta nuclear import dysfunction could be therapeutic. Thus, can the deficits observed in Figure 4 be corrected by upregulation of importin-beta activity, for example, by overexpressing importin-beta, importin-alpha, or perhaps both as has been suggested by Rossoll and colleagues (PMID: 36482422 & PMID: 38254150).

2. By the same logic, can the deficits observed in Figure 4 be exacerbated by importazole.

Addition of these studies will help confirm or refute the central hypothesis of this paper.

**Have all data underlying the figures and results presented in the manuscript been provided?**

Reviewer #1: None

Reviewer #2: None

Reviewer #3: **No: **Numerical data missing?

PLOS authors have the option to publish the peer review history of their article (what does this mean?). If published, this will include your full peer review and any attached files.

Reviewer #1: No

Reviewer #2: No

Reviewer #3: No

---

## [Decision Letter · Decision Letter 1]

28 Aug 2024

Dear Dr Sher,

We are pleased to inform you that your manuscript entitled "NEMF mutations in mice illustrate how Importin-β specific nuclear transport defects recapitulate neurodegenerative disease hallmarks" has been editorially accepted for publication in PLOS Genetics. Congratulations!

Yours sincerely,

Gregory A. Cox

Academic Editor

PLOS Genetics

Scott Williams

Section Editor

PLOS Genetics

Comments from the reviewers (if applicable):

Reviewer's Responses to Questions

**Comments to the Authors:**

Reviewer #2: The authors have substantially revised this manuscript and in doing so have sufficiently addressed my concerns.

Reviewer #3: The authors have addressed my prior concerns.

**Have all data underlying the figures and results presented in the manuscript been provided?**

Reviewer #2: None

Reviewer #3: Yes

PLOS authors have the option to publish the peer review history of their article (what does this mean?). If published, this will include your full peer review and any attached files.

Reviewer #2: No

Reviewer #3: No

**Data Deposition**

http://datadryad.org/submit?journalID=pgenetics&manu=PGENETICS-D-24-00242R1

**Press Queries**

---

## [Editor Report · Acceptance letter]

19 Sep 2024

PGENETICS-D-24-00242R1 

NEMF mutations in mice illustrate how Importin-β specific nuclear transport defects recapitulate neurodegenerative disease hallmarks 

Dear Dr Sher, 

We are pleased to inform you that your manuscript entitled "NEMF mutations in mice illustrate how Importin-β specific nuclear transport defects recapitulate neurodegenerative disease hallmarks" has been formally accepted for publication in PLOS Genetics! Your manuscript is now with our production department and you will be notified of the publication date in due course.

With kind regards,

Anita Estes

PLOS Genetics

On behalf of:
